# Optimal coding and neuronal adaptation in economic decisions

Aldo Rustichini[1], Katherine E. Conen[2], Xinying Cai[2,5] & Camillo Padoa-Schioppa [2,3,4]

During economic decisions, offer value cells in orbitofrontal cortex (OFC) encode the values of offered goods. Furthermore, their tuning functions adapt to the range of values available in any given context. A fundamental and open question is whether range adaptation is behaviorally advantageous. Here we present a theory of optimal coding for economic decisions. We propose that the representation of offer values is optimal if it ensures maximal expected payoff. In this framework, we examine offer value cells in non-human primates. We show that their responses are quasi-linear even when optimal tuning functions are highly non-linear. Most importantly, we demonstrate that for linear tuning functions range adaptation maximizes the expected payoff. Thus value coding in OFC is functionally rigid (linear tuning) but parametrically plastic (range adaptation with optimal gain). Importantly, the benefit of range adaptation outweighs the cost of functional rigidity. While generally suboptimal, linear tuning may facilitate transitive choices.

[1] Department of Economics, University of Minnesota, 1925 4th Street South 4-101, Minneapolis, MN 55455, USA. [2] Department of Neuroscience, Washington University in St Louis, 660 South Euclid Avenue, St Louis, MO 63110, USA. [3] Department of Economics, Washington University in St Louis, St Louis, MO 63130, USA. [4] Department of Biomedical Engineering, Washington University in St Louis, St Louis, MO 63130, USA. [5] Present address: NYU Shanghai, 1555 Century Ave, Room 1251, Pudong New District, Shanghai 200122, China. Correspondence and requests for materials should be addressed to C.P-S. (email: camillo@wustl.edu)

hoosing between two goods entails computing and comparing their subjective values. Evidence from lesions and neurophysiology indicates that these mental operations engage the orbitofrontal cortex (OFC)[1–3]. Experiments in which rhesus monkeys chose between different juices identified three groups of neurons in this area. Offer value cells encode the values of individual goods and are thought to provide the primary input to the decision. Conversely, chosen juice cells and chosen value cells represent the binary choice outcome and the value of the chosen good[4, 5]. The present study focuses on offer value cells.

Previous work indicated that these neurons undergo range adaptation. In any behavioral context, their firing rate is a linear function of the offered values; their tuning slope is inversely proportional to the range of values available in that context[6–8]. Prima facie, range adaptation seems to ensure an efficient neuronal representation. However, it was shown that uncorrected adaptation in offer value cells would result in arbitrary choice biases[9]—a problem conceptually analogous to the "coding catastrophe" discussed for sensory systems[10–12]. Experimental evidence presented in this study indicates that, in fact, changing the range of offer values does not affect economic preferences. In other words, range adaptation is corrected within the decision circuit to avoid choice biases. This observation raises a fundamental question: If neuronal adaptation is indeed corrected within the decision circuit, is neuronal adaptation at all beneficial to the organism? Addressing this question requires a theory of optimal coding.

Following the seminal work of Barlow[13], optimal coding has been a frequent area of research in sensory systems. A cornerstone concept is that sensory neurons are optimally tuned for perception if they transmit maximal information about the stimuli[13–15]. In any behavioral context, such optimality is achieved if tuning curves match the cumulative distribution function of the stimuli[14]. Importantly, neurons in many sensory regions adapt optimally to the current behavioral context[16–22], while tuning functions in other sensory regions seem optimized for the

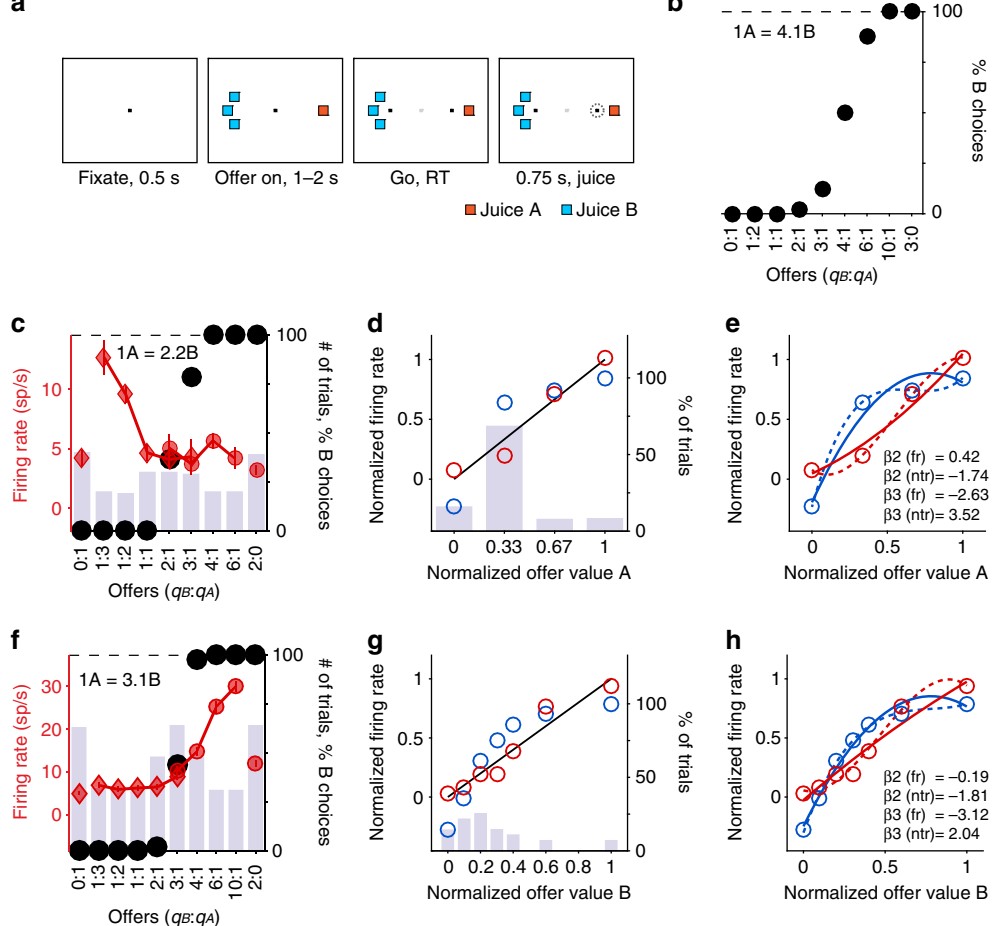

**Fig. 1** Quasi-linear coding of offer values, individual responses. **a**. Task design (see Methods). **b**. Example of choice pattern. The x-axis represents different offer types, ranked by the ratio $q_B{:}q_A$. Black dots represent the percent of "choice B" trials. **c**. Example offer value A response. Black dots represent the choice pattern. The histogram illustrates the number of trials presented for each offer type. Red symbols represent firing rates±SEM (diamonds and squares for "choice A" and "choice B", respectively). The y-axis on the left refers to firing rates. The y-axis on the right refers both to the number of trials (histogram) and to the choice pattern (black symbols). **d**. Comparing firing rates and ntrials$_{CDF}$. Same response as in **c**. The x-axis represents normalized quantity levels of juice A. The histogram illustrates the percent of trials for each quantity level. This session included 247 trials, and juice A was offered at quantity levels 0 (39 trials, 16%), 1 (169 trials, 68%), 2 (19 trials, 8%), and 3 (20 trials, 8%). Note that low quantity levels were over-represented. Blue circles represent the cumulative distribution function for the number of trials (ntrials$_{CDF}$). The y-axis on the right refers both to the number of trials (histogram) and to ntrials$_{CDF}$ (blue circles). Red circles represent firing rates. Here each neuronal data point is an average across all the trials with given quantity level (not across a single trial type). The y-axis on the left refers to normalized firing rates. Limits on the y-axes were set such that the same line (black) represents the best linear fit for firing rates and for ntrials$_{CDF}$. (Because all measures are normalized, this is the identity line.) **e**. Curvature of firing rates and ntrials$_{CDF}$. Same data points as in **d**. Continuous and dotted lines are the result of the quadratic and cubic fit, respectively. **f–h**. Example offer value B response

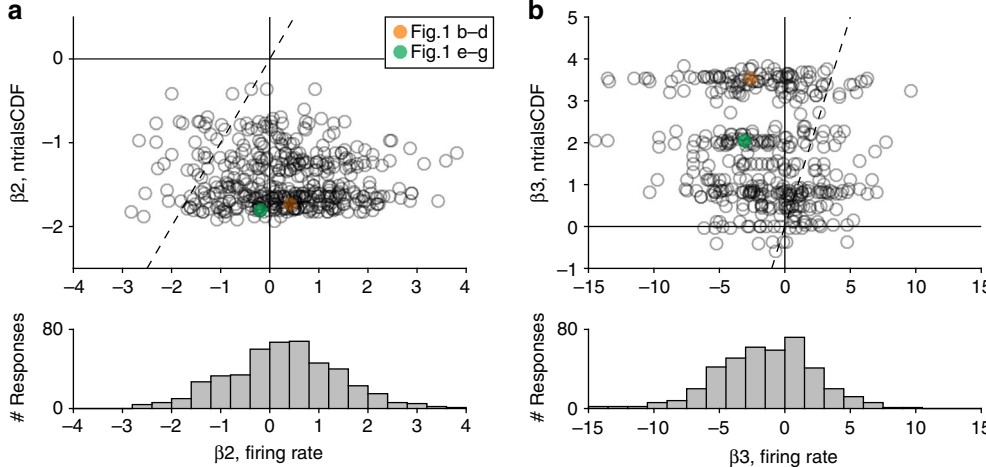

**Fig. 2** Quasi-linear coding of offer values, population analysis ($N=447$). **a** Quadratic term. Each data point in the scatter plot represents one response. The $x$-axis and $y$-axis represent $\beta_{2,\ firing\ rate}$ and $\beta_{2,\ ntrials\ CDF}$, respectively. The dotted line represents the identity line, and the responses illustrated in Fig. 1 are highlighted. The histogram recapitulates the distribution for $\beta_{2,\ firing\ rate}$. Since low offer values were always over-represented in the experiments, generally $\beta_{2,\ ntrials\ CDF} < 0$. In contrast, measures for $\beta_{2,\ firing\ rate}$ were broadly scattered above and below zero (see histogram). **b** Cubic term. Same conventions as in **a**. Generally, $\beta_{3,\ ntrials\ CDF} > 0$. In contrast, measures for $\beta_{3,\ firing\ rate}$ were broadly scattered above and below zero. Notably, on average across the population, measures for $\beta_{2,\ firing\ rate}$ were close to, but significantly above zero (mean($\beta_{2,\ firing\ rate}$) = 0.28; $p < 10^{-6}$, $t$-test). Conversely, measures for $\beta_{3,\ firing\ rate}$ were close to, but significantly below zero (mean($\beta_{3,\ firing\ rate}$) = −1.42; $p < 10^{-12}$, $t$-test). In both cases, the deviance from zero measured for $\beta_{\bullet,\ firing\ rate}$ was in the direction opposite to that observed for the corresponding $\beta_{\bullet,\ ntrials\ CDF}$, indicating that tuning functions did not match ntrials$_{CDF}$ averaged across sessions. We return to these deviances later in the article

distribution of natural stimuli[16, 23, 24]. Because they constitute the input layer of the decision circuit, offer value cells are in some ways analogous to sensory cells. One might thus wonder whether their tuning functions match the cumulative distribution function of the offered values. Experimental evidence presented here indicates that this is not the case. More specifically, we show that the tuning functions of offer value cells are quasi-linear and not correlated with the cumulative distribution function of offered values (or its average across sessions). Thus the coding of offer values in OFC, while context-adapting, is not optimal in the sense of information transmission.

In this article, we introduce a new theory of optimal coding for economic decisions. In essence, we propose that offer value neurons are optimally tuned for economic decisions if they ensure maximal expected payoff. In this framework, we present a series of theoretical and experimental results. Behavioral and neuronal data were collected in two experiments in which monkeys chose between different juices offered in variable amounts. First, assuming linear tuning functions, we demonstrate that range adaptation, corrected to avoid choice biases, ensures maximal expected payoff. Second, confirming theoretical predictions, we show that expected payoff and value range are inversely related in the experiments. Third, relaxing the assumption of linearity, we demonstrate that optimal response functions in our experiments were in fact non-linear. Hence, linearity is a rigid property of value coding not subject to contextual adaptation. Fourth, we show that the benefit afforded by range adaptation outweighs the cost imposed by functional rigidity. In other words, quasi-linear but range-adapting tuning functions are sufficient to ensure close-to-optimal choice behavior. Taken together, these results shed new light on the nature of value coding and the role played by neuronal adaptation in economic decisions.

## Results

**Relative value, choice variability and expected payoff.** In Exp. 1, monkeys chose between two juices (A and B, with A preferred) offered in variable amounts (Fig. 1a, b). The range of quantities offered for each juice remained fixed within a session, while the

quantity offered on any given trial varied pseudo-randomly. Monkeys' choices generally presented a quality–quantity trade-off. If the two juices were offered in equal amounts, the animal would generally choose A (by definition). However, if sufficiently large quantities of juice B were offered against one drop of juice A, the animal would choose B. The "choice pattern" was defined as the percentage of trials in which the animal chose juice B as a function of the offer type. In each session, the choice pattern was fitted with a sigmoid function, and the flex of the sigmoid provided a measure for the relative value of the two juices, referred to as $\rho$ (see Methods). The relative value allows one to express quantities of the two juices on a common value scale. In one representative session, we measured $\rho = 4.1$ (Fig. 1b).

Choice patterns often presented some variability. For example, consider in Fig. 1b offers 6B:1A. In most trials, the animal chose juice B, consistent with the fact that the value of 6B was higher than the value of 1A. However, in some trials, the animal chose the option with the lower value. Similarly, in some trials, the animal chose 3B over 1A. Intuitively, choice variability is high when the sigmoid is shallow. Thus in each session, the steepness of the fitted sigmoid, referred to as $\eta$, quantified the (inverse of) choice variability (see Methods).

In any given trial, we define the payoff as the value chosen by the animal. Thus given a set of offers and a sigmoid function, the expected payoff is equal to the chosen value averaged across trials. Importantly, the expected payoff is inversely related to choice variability, and thus directly related to the steepness of the sigmoid. When the sigmoid is steeper, choice variability is lower, and the expected payoff is higher; when the sigmoid is shallower, choice variability is higher, and the expected payoff is lower. Notably, the relative value of two juices is entirely subjective. In contrast, a key aspect of the expected payoff is objective: given a set of offers, a relative value and two sigmoid functions, the steeper sigmoid yields higher expected payoff.

**Quasi-linear coding of offer values.** While animals performed the task, we recorded the activity of individual neurons in the central OFC. Firing rates were analyzed in multiple time

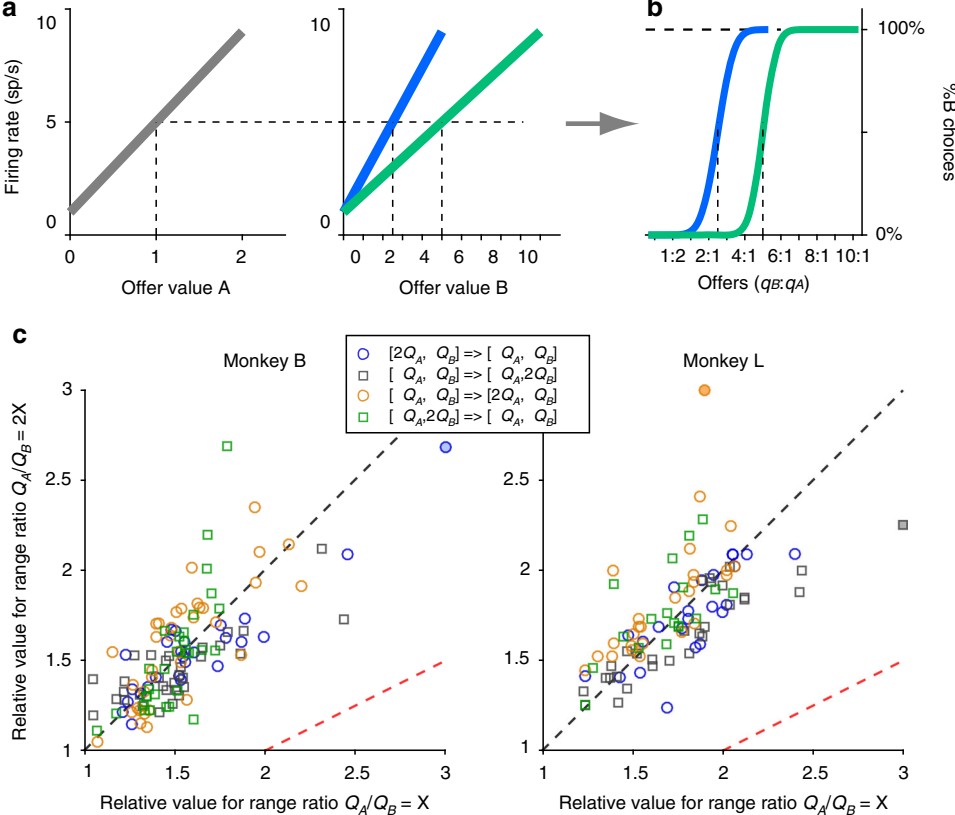

**Fig. 3** Range adaptation is corrected within the decision circuit. **a**, **b** Uncorrected range adaptation would induce arbitrary choice biases. Panel **a** shows the schematic response functions of two neurons encoding the offer value A (left) and the offer value B (right). Panel **b** shows the resulting choice patterns under the assumption that decisions are made by comparing the firing rates of these two cells. We consider choices in two conditions, with the range $\Delta A$ = [0 2] kept constant. When $\Delta B$ = [0 5], the firing rate elicited by offer 1 A is between that elicited by offers 2B and 3B ($\rho$ = 2.5). When $\Delta B$ = [0 10], offer value B cells adapt to the new value range. Now offer 1 A elicits the same firing rate as offer 5B ($\rho$ = 5). Thus if range adaptation is not corrected, changing either value range induces a choice bias. Importantly, this issue would vanish if both neurons adapted to the same value range. However, experimental evidence indicated that each population of offer value cells adapts to its own value range [9]. **c** Relative values measured in Exp.2. The two panels refer to the two animals. In each panel, the axes represent the relative value measured when $Q_A/Q_B$ = X (x-axis) and that measured when $Q_A/Q_B$ = 2X (y-axis). Each data point represents data from one session, and different symbols indicate different protocols (see legend). If decisions were made by comparing uncorrected firing rates, data points would lie along the red dotted line. In contrast, data points lie along the black dotted line (identity line). In other words, the relative values measured in the two trial blocks were generally very similar, indicating that range adaptation was corrected within the decision circuit. Panels **a** and **b** are reproduced from[9]

windows. In each session, an "offer type" was defined by a pair of offers (e.g., [1A:3B]); a "trial type" was defined by an offer type and a choice (e.g., [1A:3B, B]); a "neuronal response" was defined as the activity of one cell in one time window as a function of the trial type. Earlier work showed that different responses encoded variables offer value, chosen value and chosen juice[4, 5]. Unless otherwise indicated, the present analyses focus on offer value responses.

Previous studies failed to emphasize that the tuning of offer value cells was quasi-linear even though the distribution of values was highly non-uniform. To illustrate this point, we identified for each offer value response the quantity levels for the corresponding juice, and we calculated the number of trials in which each quantity level had been presented to the animal within the session. For example, Fig. 1c illustrates one offer value A response. In this session, juice A was offered in quantity levels (number of trials): 0 (39), 1 (169), 2 (19), and 3 (20). For each set of trials, we computed the mean firing rate of the cell. In addition, we computed the cumulative distribution function for the number of trials ($ntrials_{CDF}$) as a function of the quantity level. By analogy with sensory systems[14], neurons encoding $ntrials_{CDF}$ would provide maximal information about the offer values. Firing

rates and $ntrials_{CDF}$ were highly correlated: for both of them, a linear regression on the quantity level provided a reasonably good fit (Fig. 1d). However, the non-uniform distribution of offer values induced a curvature in $ntrials_{CDF}$. Similarly, each neuronal response taken alone always presented some curvature. To assess whether and how the curvature in neuronal responses was related to the curvature in $ntrials_{CDF}$, we normalized both firing rates and $ntrials_{CDF}$ (see Methods). We thus fit each set of data points with a 2D polynomial, which provided a coefficient for the quadratic term ($\beta_2$). Separately, we fit each set of data points with a 3D polynomial, which provided a coefficient for the cubic term ($\beta_3$; Fig. 1e; see also Fig. 1f–h).

Because lower offer values were over-represented in the experiments, we generally measured $\beta_{2, ntrialsCDF} < 0$ and $\beta_{3, ntrialsCDF} > 0$. In contrast, $\beta_{2, firing rate}$, and $\beta_{3, firing rate}$ varied broadly across the population, and their distributions were fairly symmetric around zero (Fig. 2a, b). In other words, neuronal response functions were, on average, quasi-linear. These results held true for individual monkeys, in each time window, and independently of the sign of the encoding (Supplementary Fig. 1). Similar results were also obtained for chosen value responses (Supplementary Fig. 2).

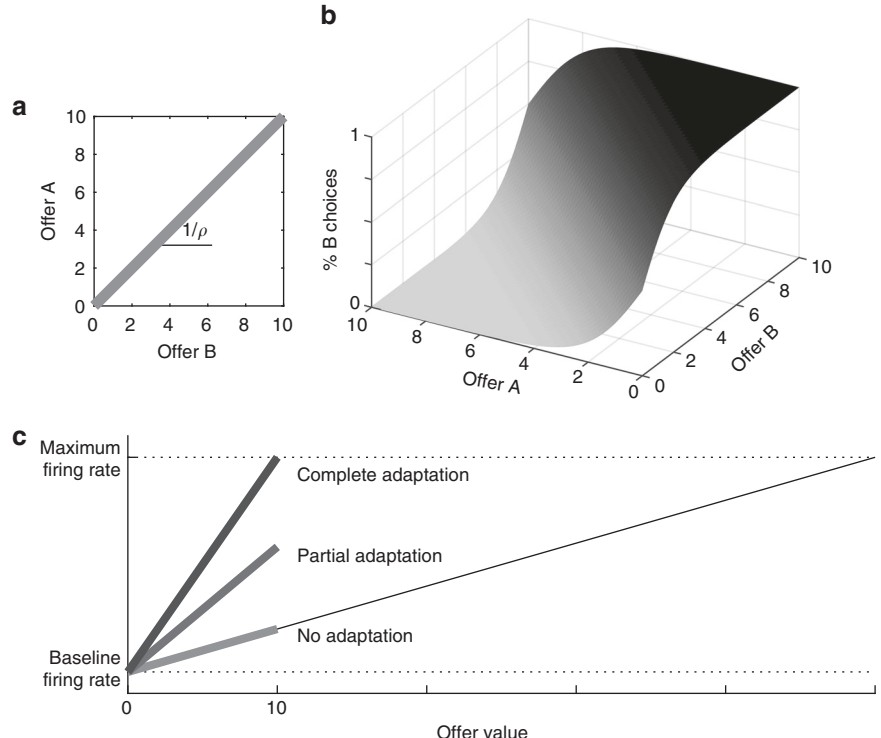

**Fig. 4** Possible adaptation scenarios. **a** Indifference line. We indicate with $q_A$ and $q_B$ the quantities of good A and good B, respectively. Across trials, $q_A$ varies in the range $[0, Q_A]$, while $q_B$ varies in the range $[0, Q_B]$. In the plane defined by $q_A$ and $q_B$, we define the "indifference curve" as the set of offers for which the animal splits decisions equally between the two goods. We assume that the indifference curve is a straight line on this plane. Thus the relative value between the two goods, referred to as $\rho$, is defined by the slope of the indifference line (slope $= 1/\rho$). **b** Choice pattern. Given offers of goods A and B, a choice pattern can be represented as a sigmoid surface, in which the z-axis represents the likelihood of choosing good B. For each pair of offers, one of the two options provides a higher payoff, depending on whether it is above or below the indifference line. However, unless the sigmoid is a step function, in some trials the animal fails to choose that option (choice variability). Given a set of offers and a relative value, the expected payoff is an increasing function of the sigmoid steepness. **c**. Adaptation scenarios. In this cartoon, offer values in the current context vary in the range [0 10]. The light line represents a hypothetical scenario in which there is no range adaptation (see Results). The darker lines represent the scenarios with partial and complete range adaptation

**Range adaptation is corrected within the decision circuit.** As previously shown, offer value cells undergo range adaptation (Supplementary Fig. 3a–c)[6]. Linear tuning implies that any given value interval is allotted the same activity interval in the neuronal representation. Range adaptation ensures that the full activity range is always available to represent the range of values offered in the current context. Thus range adaptation seems to provide an efficient representation for offer values. However, range adaptation also poses a computational puzzle[9] illustrated in Fig. 3a, b. In essence, current models assume that binary economic decisions are made by comparing the firing rates of two neuronal populations encoding the subjective values of the offered goods[25–29]. If so, by varying the ranges of the two offers one could impose any indifference point (an arbitrary choice bias).

Exp.2 was conducted to test this prediction in controlled conditions. In each session, monkeys chose between two juices. Trials were divided in two blocks. Across blocks, we either halved or doubled the range of one of the two juices ($2 \times 2$ design). For each trial block, $Q_A$ and $Q_B$ indicate the maximum quantities of juices A and B offered, respectively. Thus independently of other factors, the ratio $Q_A/Q_B$ changed by a factor of two between blocks ($Q_A/Q_B = X$ or 2X). The experimental design controlled for juice-specific satiety and other possible sources of choice bias (see Methods).

We collected behavioral data in 220 sessions. In each session and each trial block, we measured the relative value of the juices. We then compared the measures obtained in the two trial blocks.

According to the argument in Fig. 3a, b, the relative value measured when $Q_A/Q_B = X$ should be roughly twice that measured when $Q_A/Q_B = 2X$. Contrary to this prediction, the relative values measured in the two trial blocks were generally similar (Fig. 3c). Pooling all sessions, the ratio of relative values measured for the two trial blocks was statistically indistinguishable from 1 (mean ratio = 1.006; $p = 0.81$, Wilcoxon signed rank test) and significantly below 2 ($p < 10^{-37}$, Wilcoxon signed rank test). These results held true for each animal.

**Range adaptation maximizes the expected payoff.** Exp. 2 indicated that range adaptation is corrected within the decision circuit. We previously proposed a possible scheme for this correction. In essence, choice biases are avoided if the synaptic efficacies between offer value cells and downstream neuronal populations are rescaled by the value ranges[9, 29]. However, if this correction occurs, it is reasonable to question whether range adaptation benefits the decision process at all. The central result of this study is that range adaptation in offer value cells maximizes the expected payoff even if adaptation is corrected within the decision circuit. The theoretical argument is summarized here and detailed in the Supplementary Note, where we provide mathematical proofs.

Consider the general problem of choices between two goods, A and B. We indicate the quantities of A and B offered on a particular trial with $q_A$ and $q_B$. Across trials, $q_A$ varies in the range

$[0, Q_A]$, while $q_B$ varies in the range $[0, Q_B]$. We assume linear indifference curves (Fig. 4a) and indicate the relative value with $\rho$. Choices can be described by a sigmoid surface (Fig. 4b). For each pair of offers, one of the two options provides a higher payoff, but in some trials the animal fails to choose that option (choice variability). Intuitively, this may happen because the neural decision circuit has a finite number of neurons, limited firing rates, trial-by-trial variability in the activity of each cell, and non-zero noise correlations.

Figure 4c illustrates the issue of interest. We assume that neuronal response functions are linear. Actual neurons always have a baseline firing rate (corresponding to a zero offer), but we assume that this activity does not contribute to the decision. Thus we focus on baseline-subtracted response functions. Let us consider a hypothetical scenario in which there is no adaptation. If so, neurons would have fixed tuning, corresponding to a linear response function defined on a very large value range. In contexts where the encoded good varies on a smaller range, neuronal firing rates would span only a subset of their potential activity range. In contrast, if neurons undergo complete range adaptation, firing rates span the full activity range in each behavioral context.

To understand how range adaptation in offer value cells affects the expected payoff, it is necessary to consider a specific decision model. That is, the question must be addressed under some hypothesis of how the activity of offer value cells is transformed into a decision. We examined the linear decision model[30, 31] formulated as follows:

$$D = X^A - X^B$$
$$X^g = K_g \sum_i w_i^g r_i^g \qquad g = A, B \quad i = 1 \dots n \qquad (1)$$

where $r_i^g$ is the firing rate of an offer value $g$ cell, $w_i^g$ are decision weights, $n$ is the number of cells associated with each juice, and $K_g$ is the synaptic efficacy of offer value $g$ cells onto downstream populations. Conditions $D > 0$ and $D < 0$ correspond to choices of goods A and B, respectively.

We model the firing rates of offer value cells as Poisson variables and we approximate noise correlations with their mean long-distance component[30]. In accord with experimental measures, we set the noise correlation to $\xi = 0.01$ for pairs of neurons associated with the same good, and to zero for pairs of neurons associated with different goods[30]. Importantly, $\xi$ does not depend on firing rates (Supplementary Fig. 4). We thus compute the probability of choosing juice A given offers $q = (q_A, q_B)$, tuning slopes $t = (t_A, t_B)$ and synaptic efficacies $K = (K_A, K_B)$:

$$P(ch = A|q, t, K) = \Pr\left( Z \geq -\frac{K_A q_A t_A - K_B q_B t_B}{\sqrt{\chi(K_A^2 q_A t_A + K_B^2 q_B t_B)}} \middle| Z \sim N(0, 1) \right) \qquad (2)$$

where $N(0, 1)$ is the standard normal distribution and $\chi = \xi/4$.

Eq. 2 allows one to calculate the expected payoff. Indicating with $\overline{\nu}$ the maximum possible firing rate, we demonstrate that the expected payoff is maximal when $t_g = \overline{\nu}/Q_g$. This condition corresponds to complete range adaptation (Fig. 4c). In the symmetric case, defined by $\rho Q_A = Q_B$ (equal value ranges), the expected payoff is maximal when $K_A/K_B = 1$ and there is no choice bias. In the general, asymmetric case (unequal value ranges), the expected payoff is maximal when $K_A/K_B \approx \rho Q_A/Q_B$. In this condition, there is a small choice bias that favors the larger value range and depends on $\chi$.

Eq. 2 expresses the sigmoid surface describing choices. By computing the slope of this surface on the indifference line, we show that under optimal coding the steepness of the sigmoid is inversely related to the value ranges (Supplementary Note, Eq. 28).

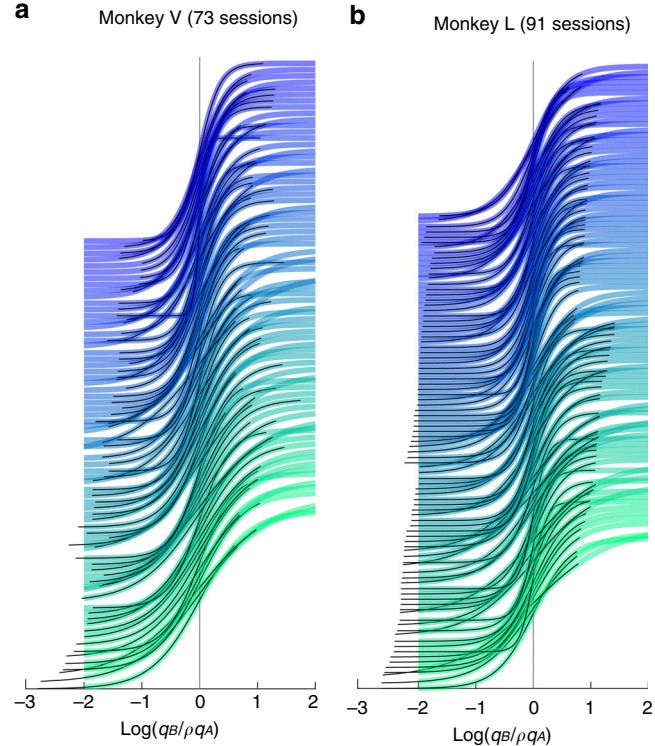

**Fig. 5** Relation between sigmoid steepness and value range. **a** Monkey V (73 sessions). For each session, the sigmoid fit provided measures for $\rho$ and $\eta$ (Eq. 6), and we computed the geometric mean value range $\Delta$. In this plot, different sigmoid functions are aligned at the flex ($x$-axis) and ranked based on $\Delta$, from top (small $\Delta$) to bottom (large $\Delta$). For each sigmoid, the thick colored line (blue-green) depicts the result of the fit in a standard interval [−2 2]. The thin black line highlights the range of values actually used in the corresponding session. Different shades of color (from blue to green) indicate the ordinal ranking of sessions according to $\Delta$. Notably, sigmoid functions at the bottom of the figure (larger $\Delta$) were shallower (lower $\eta$). **b** Monkey L (91 sessions). Same format as in **a**

**Relation between choice variability and value range.** The previous section summarizes a theory of optimal coding of offer values for economic decisions. The main prediction for linear response functions is that the slope of the encoding should be inversely proportional to the value range, as is indeed observed in the experiments (range adaptation; Supplementary Fig. 3d, e). The theory also makes another testable prediction. Consider experiments in which monkeys choose between two juices and value ranges vary from session to session. The sigmoid steepness should decrease as a function of the value ranges. To test this prediction, we examined 164 sessions from Exp.1. For each session, we computed the geometric mean value range $\Delta \equiv (\rho Q_A Q_B)^{1/2}$, and we obtained a measure for the sigmoid steepness ($\eta$) from the sigmoid fit. We thus examined the relation between $\eta$ and $\Delta$.

Figure 5a, b illustrates the fitted sigmoid obtained for each experimental session in our data set, separately for monkeys V and L. For each animal, sigmoid functions were aligned at the flex and ranked according to $\Delta$. Notably, sigmoid functions with small $\Delta$ were generally steeper (large $\eta$), while sigmoid functions with large $\Delta$ were generally shallower (small $\eta$). In other words, there was a negative correlation between $\eta$ and $\Delta$. This correlation, summarized in a scatter plot (Fig. 6), was statistically significant in each animal (monkey V: corr coef $= -0.41$, $p < 0.0005$; monkey L: corr coef $= -0.26$, $p < 0.02$). Control analyses confirmed that this result was not due to differences between juice pairings (Supplementary Fig. 5) or to fluctuations in the relative

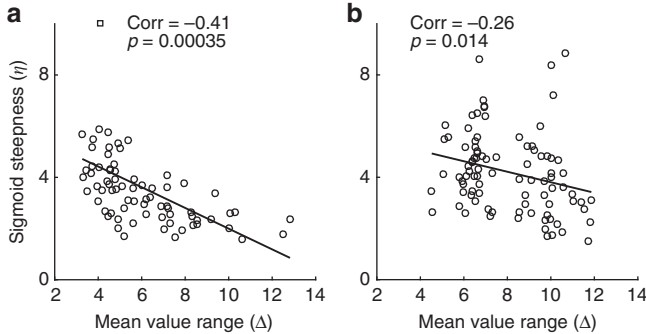

**Fig. 6** Relation between sigmoid steepness and value range, scatter plots. **a**, **b**. Panels **a** and **b** refer to monkey V (73 sessions) and monkey L (91 sessions), respectively. In each panel, the x-axis represents the geometric mean value range $\Delta \equiv (\rho\, Q_A\, Q_B)^{1/2}$, the y-axis represents the steepness of the sigmoid ($\eta$) and each data point represents one session. In both animals, the two measures were significantly and negatively correlated. In each panel, the black line represents the result of Deming's regression (see Methods)

value (Supplementary Fig. 6). Similar results were also obtained for data from Exp.2 (Supplementary Fig. 7).

**Neuronal responses are functionally rigid.** We have shown that range adaptation maximizes the expected payoff under the assumption of linear response functions. Next we address a closely related question, namely whether (or in what sense) linear response functions are optimal in the first place. In the visual system, optimal coding is achieved if tuning functions match the cumulative distribution of the encoded stimuli[14], [19]. In the valuation system, the equivalent condition would occur if offer value responses matched the cumulative distribution of offered values. We already showed that this is not the case (Fig. 2). In retrospect, this finding is not surprising because a subject performing economic decisions is best served by response functions that maximize the expected payoff, which do not necessarily maximize information transmission. Thus what is the optimal response function for offer value cells?

The answer to this question depends on the joint distribution of offers and on the relative value of the two goods. For example, consider the case in which an animal chooses between goods A and B and $\rho = 2$. Good A is always offered in quantity 1, while good B is offered in quantities between 0 and 5 (Fig. 7a). We consider offer value B cells and we indicate with $r_B$ their firing rate. It is easy to see that the payoff is maximal if $r_B(x) = 0$ when $x < 2$, $r_B(2) = 0.5$, and $r_B(x) = 1$ when $x > 2$, where x are quantities of juice B offered. Hence, the optimal response function is a step function with the step located at $x = 2$. Next consider the case in which quantities of both goods vary between 0 and 5, at least one of the two goods is always offered in quantity 1, and $\rho = 2$ (Fig. 7b). Again, the optimal response function for offer value B cells is $r_B(x) = 0$ when $x < 2$, $r_B(2) = 0.5$, and $r_B(x) = 1$ when $x > 2$. For offer value A cells, the optimal response function is $r_A(0) = 0$, $r_A(1) = 0.5$, and $r_A(x) = 1$ when $x > 1$. Thus for both goods, the optimal response function is a step function. Analogously, if offer types are the same but $\rho = 3$ (Fig. 7c), the optimal response function for offer value B cells is a step function with the step located at $x = 3$.

The scenarios depicted in Fig. 7b, c are similar to those occurring in Exp.1. Indeed our sessions always included forced choices for both juices. Furthermore, in 96% (200 out of 208) of our sessions, when both juices were offered, at least one of them was offered in quantity 1 (Supplementary Fig. 8). Thus in Exp.1, optimal response functions for offer value cells would have been

step functions, not linear functions. Our neuronal data clearly belied this prediction (Fig. 2). In other words, our results indicate that the functional form of offer value cells did not adapt to maximize the payoff in each session. To further examine this point, we ran two additional analyses.

First, we entertained the hypothesis that the functional form of offer value cells might adapt on a longer time scale, over many sessions. However, we found that the mean optimal response function was a fairly sharp sigmoid (Fig. 7d), contrary to our observations (Fig. 2). Second and most important, we recognized that neuronal responses examined in Fig. 2 were originally identified through a variable selection analysis that only considered linear response functions[4] (see Methods). This effectively imposed a bias in favor of linearity. To eliminate this bias, we repeated the variable selection procedures including in the analysis all the variables discussed in this study. These included the cumulative distribution function of offer values ($ntrials_{CDF}$), the optimal responses in each session (step functions) and the mean optimal response function across sessions (Methods). The results confirmed previous findings: variables offer value, chosen value and chosen juice still provided the highest explanatory power. In particular, the explanatory power of linear offer value variables was significantly higher than that of each of the new variables (Supplementary Table 1).

In the final analysis of this section, we considered whether the response functions observed experimentally would maximize the expected payoff for other possible joint distributions of offers. To do so, we generalized the theory of optimal coding by relaxing the assumption of linear response functions (Supplementary Note, Section 6). One interesting candidate was the symmetric uniform distribution (Fig. 7e). We calculated the optimal response functions given this distribution ($ORF_{uniform}$) and we found that they are quasi-linear and slightly convex (Fig. 7e). Notably, this non-linearity is in the same direction observed in Fig. 2a (histogram). We then repeated the variable selection analysis including variables based on $ORF_{uniform}$. Interestingly, neuronal responses best explained by $ORF_{uniform}$ variables were more numerous than those best explained by linear offer value variables (Fig. 8). As in previous studies[4], we used two procedures for variable selection, namely stepwise and best-subset (Methods). Both procedures identified variables offer A $ORF_{uniform}$, offer B $ORF_{uniform}$, chosen value and chosen juice as providing the maximum explanatory power (Fig. 9). However, a post-hoc analysis indicated that the explanatory power of $ORF_{uniform}$ variables was statistically indistinguishable from that of linear offer value variables (Supplementary Table 2).

In conclusion, the variable selection analyses confirmed that offer value responses were quasi-linear and thus suboptimal given the joint distributions of offers in our experiments. Furthermore, offer value responses were indistinguishable from optimal responses functions calculated assuming a uniform joint distribution of offers. We elaborate on the significance of this finding in the Discussion.

**Cost of functional rigidity and benefit of range adaptation.** The tuning of offer value cells is functionally rigid (quasi-linear) but parametrically plastic (range adapting with optimal gain). In terms of the expected payoff, functional rigidity ultimately imposes some cost, while range adaptation ultimately yields some benefit. We sought to quantify these two terms in our experiments.

For each session of Exp.1, we focused on strictly binary choices (i.e., we excluded forced choices). On the basis of the relative value of the juices ($\rho$), we computed for each trial the chosen value (i.e., the payoff) and the max value, defined as the higher of

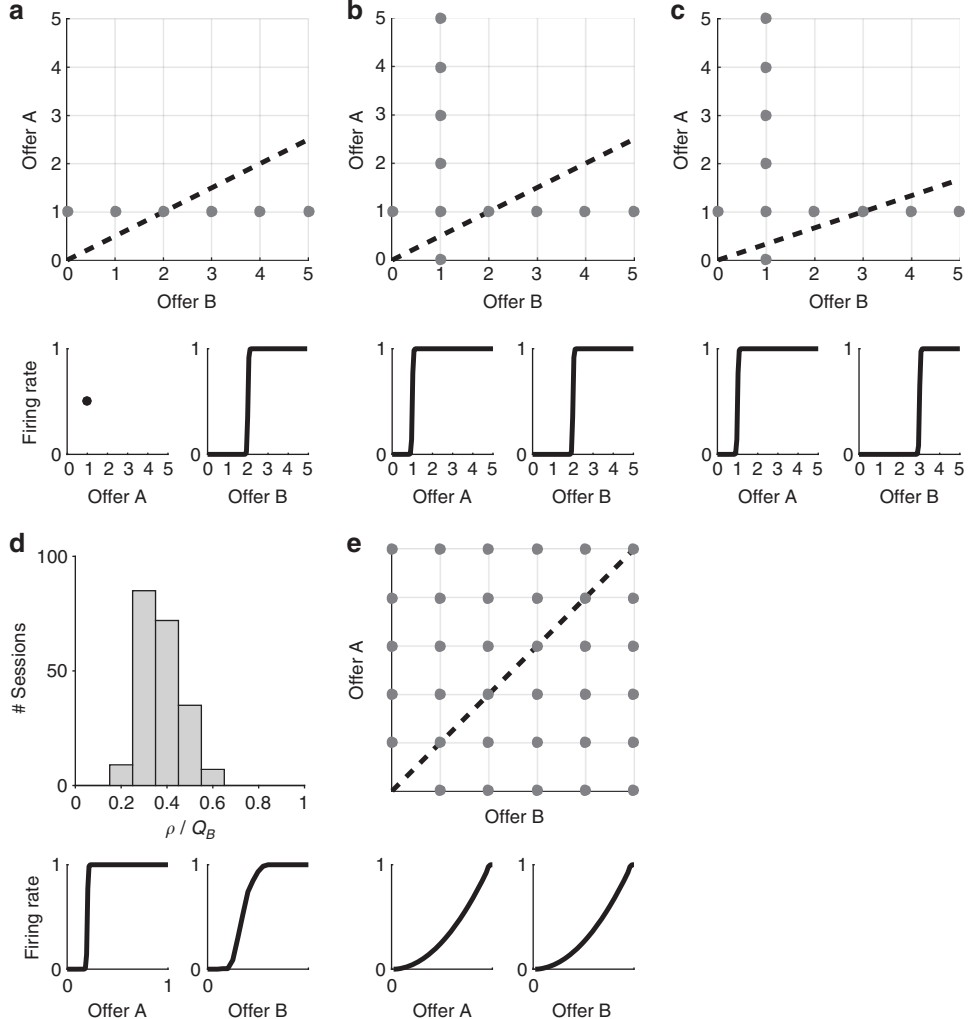

**Fig. 7** Optimal response functions. **a** One good offered in fixed quantity ($\rho = 2$). Gray dots represent offer types presented in the session and the dotted line represents the indifference line. Good A is always offered in quantity 1 while good B varies in the range [0 5]. Optimal response functions are shown in the lower panels. **b** Idealized experimental session ($\rho = 2$). For each good, quantities vary in the range [0 5], but in each offer type at least one good is offered in quantity 1. Lower panels show the optimal response functions (ORF, step functions). **c** Idealized experimental session ($\rho = 3$). **d** Optimal mean response functions. The histogram represents the distribution of $\rho/Q_B$, where $\rho$ is the relative value and $Q_B$ is the maximum quantity of juice B offered. Lower panels show the mean optimal response functions, mean(ORF). For offer value B, the response function is computed as the cumulative distribution function for $\rho/Q_B$. **e** Idealized session with uniform distribution and equal value ranges (a.u.). Lower panels show the corresponding optimal response functions (ORF$_{uniform}$). Note that the curvature of ORF$_{uniform}$ is in the same direction as that observed on average in the neuronal population (Fig. 2a, histogram)

the two values offered in that trial. We also defined the chosen value$_{chance}$ as the chosen value expected if the animal chose randomly between the two offers. Hence, chosen value$_{chance}$ = (offer value A+offer value B)/2. For each session we defined the fractional lost value (FLV) as:

$$\text{FLV} = \text{fractional lost value} = \frac{\langle \text{max value} - \text{chosen value} \rangle}{\langle \text{max value} - \text{chosen value}_{chance} \rangle}$$
(3)

where brackets indicate an average across trials. Under normal circumstances, FLV varies between 0 and 1. Specifically, FLV = 0 if the animal always chooses the higher value (chosen value = max value) and FLV = 1 if the animal always chooses randomly (chosen value = chosen value$_{chance}$). Thus FLV quantifies the fraction of value lost to choice variability. For each session, we also computed the percent error, defined as the percent of trials in

which the animal chose the lower value. We examined these metrics across sessions.

The percent error varied substantially from session to session, between 0 and 23% (Fig. 10a). On average across sessions, mean (percent error)=8.7%. The FLV also varied substantially across sessions, between 0 and 0.24 (Fig. 10b). On average across sessions, mean(FLV)=0.05. Importantly, this estimate provides an upper bound for the value lost by the animal due to suboptimal tuning functions, because other factors might also contribute to choice variability. Hence, the cost of functional rigidity in the coding of offer values may be quantified as $\leq 0.05$.

Because we cannot observe decisions in the absence of neuronal adaptation, quantifying the benefits of range adaptation requires a simulation. We proceeded as follows. For each session and for each trial, the sigmoid fit provided the probability that the animal would choose juice B ($P_{ch=B}$; see Eq. 5) or juice A ($P_{ch=A}=1-P_{ch=B}$). Thus in each trial the expected chosen value

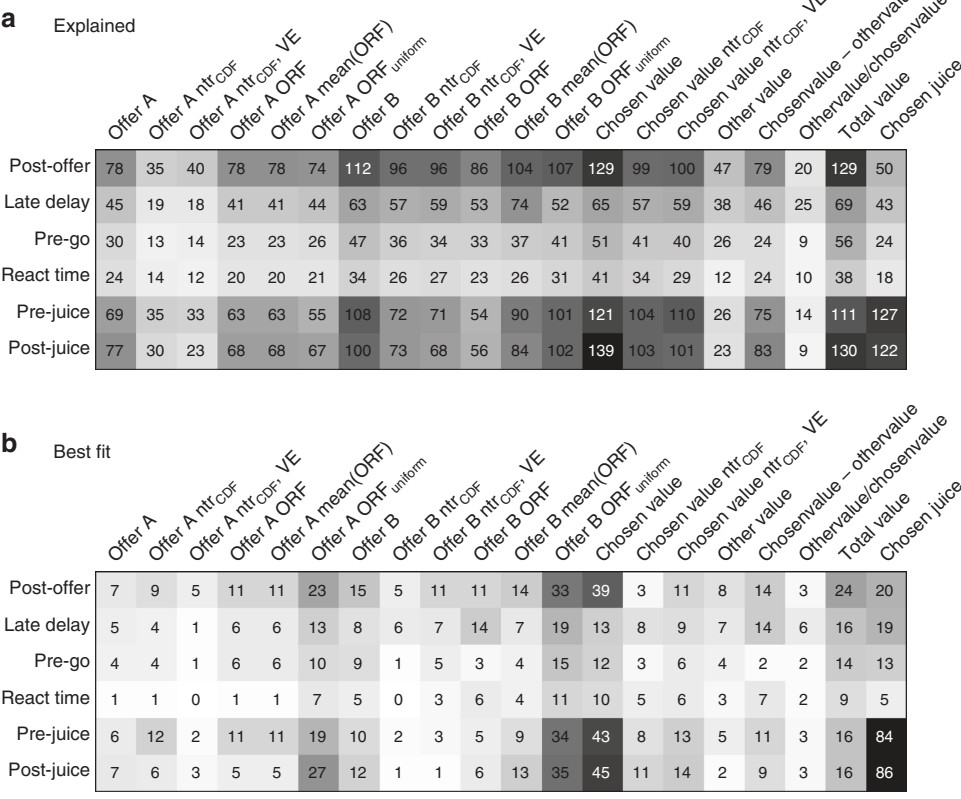

**Fig. 8** Population analysis of neuronal responses. Each neuronal response that passed an ANOVA criterion was regressed against each variable. If the regression slope differed significantly from zero, the variable was said to explain the response (see Methods). **a** Explained responses. Rows and columns represent time windows and variables, respectively. In each location, the number indicates the number of responses explained by the corresponding variable. For example, in the post-offer time window, the variable offer A (linear response function) explained 78 responses. The same numbers are also represented in grayscale. Each response could be explained by more than one variable. Thus each response might contribute to multiple bins in this panel. **b** Best fit. In each location, the number indicates the number of responses for which the corresponding variable provided the best fit (highest $R^2$). For example, in the post-offer time window, offer A (linear response function) provided the best fit for 7 responses. The same numbers are also represented in grayscale. In this panel, each neuronal response contributes at most to one bin. Qualitatively, the dominant variables appear to be offer A ORF$_{uniform}$, offer B ORF$_{uniform}$, chosen value and chosen juice. Indeed the variable selection procedures identified these variables as the ones with the highest explanatory power (Fig. 9)

(i.e., the expected payoff) was:

$$E(\text{chosen value}) = P_{ch=A} \text{ offer value A} + P_{ch=B} \text{ offer value B} \quad (4)$$

For each session, we computed the expected fractional lost value (EFLV) by substituting the E(chosen value) for the chosen value in Eq. 3. Importantly, we verified that EFLV provided a good estimate for the actual FLV (Fig. 10c).

To address the question of interest, we reasoned along the lines of Fig. 4c, where the absence of adaptation is approximated with a scenario in which neurons adapt to a very large value range. We already showed that increasing the value range decreases the sigmoid steepness (Fig. 6). Thus we examined how reducing the sigmoid steepness would affect the EFLV. We found that the effects were large. For example, when we halved the sigmoid steepness ($\eta \to \eta/2$), we obtained mean(EFLV) = 0.15; when we divided the sigmoid steepness by ten ($\eta \to \eta/10$), we obtained mean(EFLV) = 0.55 (Fig. 10d). Hence, the benefit of range adaptation, while difficult to quantify exactly, is clearly very high.

To summarize, the benefit of range adaptation outweighs the cost of functional rigidity. Our analyses suggest that a quasi-linear but range-adapting coding of offer values is sufficient to ensure close-to-optimal choice behavior.

## Discussion

Sensory neurons are optimally tuned for perception if they transmit maximal information about the stimuli. In contrast, offer value neurons are optimally tuned for economic decisions if they ensure maximal expected payoff. In this framework, we examined the activity of offer value cells in OFC. These neurons are believed to provide the primary input for economic decisions. We showed that their tuning is functionally rigid (linear responses) but parametrically plastic (range adaptation with optimal gain). We also showed that range adaptation is corrected within the decision circuit to avoid arbitrary choice biases. Critically, range adaptation ensures optimal tuning even considering this correction. Confirming theoretical predictions, we showed that choice variability is directly related to the range of values offered in any behavioral context. Finally, we showed that the benefit of range adaptation outweighs the cost of functional rigidity. Importantly, our theoretical results were derived using a linear decision model (Eq. 1)[30, 31]. Future work should extend this analysis to other decision models[25, 27, 29].

On average, offer value responses presented a small but significant departure from linearity (Fig. 2). Their convexity closely resembled that predicted for optimal response functions under a uniform joint distribution (Fig. 7e), although in a direct comparison the explanatory power of ORF$_{uniform}$ functions was not significantly higher than that of strictly linear functions. Thus

future work should address this point and consider other joint distributions that might explain neuronal responses in OFC. Nonetheless, the quasi-linear nature of value coding in OFC is noteworthy. We previously showed that the activity of neurons associated with one good does not depend on the identity or value of the other good offered in the same trial[32]. With respect to range adaptation, we also showed that each neuron adapts to its own value range, independently of the range of values offered for the other good[9]. One implication of linear responses (or optimal response functions under a symmetric uniform distribution) is that the activity of neurons associated with one particular good does not depend on the distribution of values offered for the other good, or on the relative value of the two goods. Thus quasi-linearity can be seen as yet another way in which neurons associated with one particular good are blind to every aspect of the other good. This blindness, termed menu invariance, guarantees preference transitivity[33, 34], which is a fundamental trait of economic behavior. It is tempting to speculate that quasi-linear coding might have been selected in the course of evolution because it facilitates transitive choices.

Adaptive coding has been observed in numerous brain regions that represent value-related variables including the amygdala[35], anterior cingulate cortex[36], and dopamine cells[37–39]. Independently of the specific contribution of each area to behavior, adaptation necessarily poses computational challenges analogous to the coding catastrophe discussed in sensory systems[9–11]. In essence, uncorrected adaptation makes firing rates intrinsically ambiguous. Thus neuronal adaptation at any processing stage must be corrected at later stages[40] or ultimately results in impaired behavioral performance[12, 41]. With respect to offer value cells in OFC, we previously proposed that choice biases potentially introduced by range adaptation are corrected in the synapses between these neurons and downstream populations[9, 29]. The theory of optimal coding developed here makes this same prediction, which should be tested in future experiments. Interestingly, framing[42, 43] and anchoring[44] effects documented in behavioral economics qualitatively resemble adaptation-driven choice biases, although they are quantitatively more modest. In principle, these effects could be explained if synaptic rescaling trailed neuronal range adaptation. Similar mechanisms have been hypothesized in the visual system to explain illusions and aftereffects[11, 12].

The rationale for this study rests on the assumption that offer value cells in OFC provide the primary input for the neural circuit that generates economic decisions. Support for this assumption comes from lesion studies[45–47], from the joint analysis of choice probability and noise correlation[30] and from the relation between choice variability and value range shown here. Indeed, current neuro-computational models of economic decisions embrace this view[26, 29, 48–51]. However, causal links between the activity of offer value cells and the decision have not yet been demonstrated with the gold-standard approach of biasing choices using electrical or optical stimulation. Future work should fill this important gap.

## Methods

**Experimental procedures**. All experimental procedures conformed to the NIH Guide for the Care and Use of Laboratory Animals and were approved by the

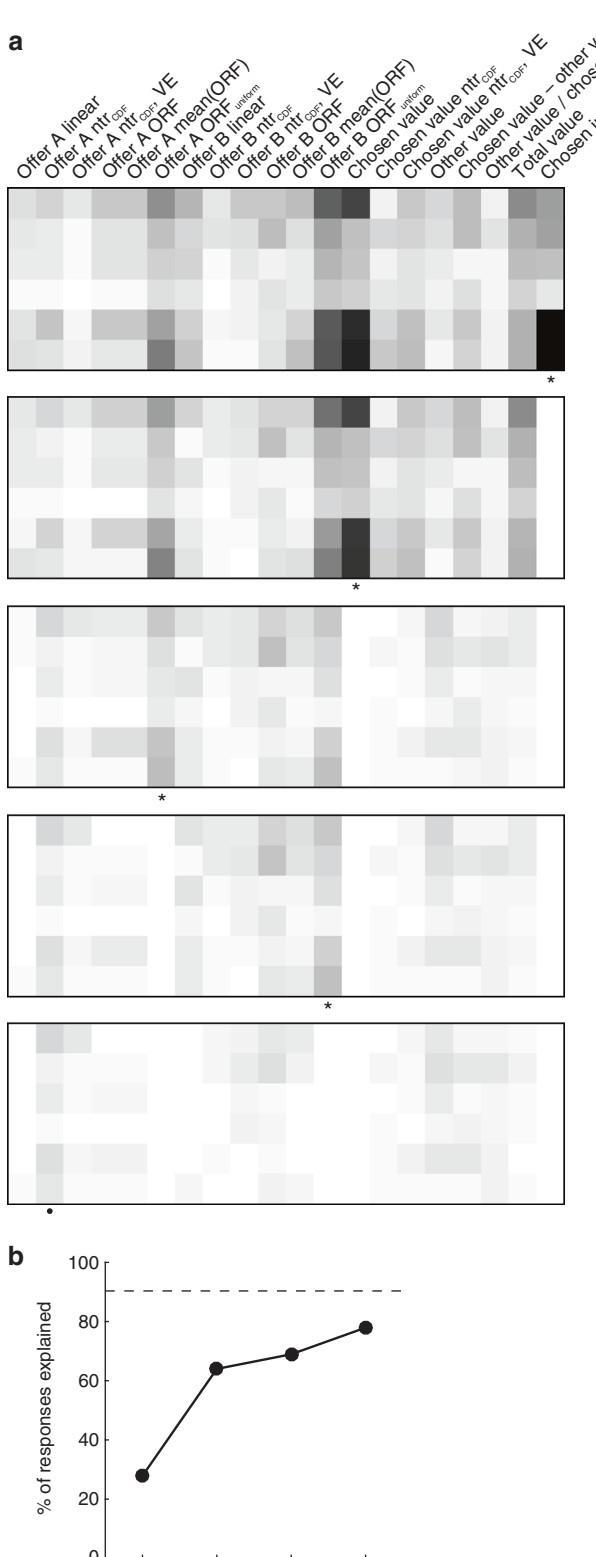

**Fig. 9** Variable selection analysis. **a** Stepwise selection. The top panel is as in Fig. 8b. At each iteration, the variable providing the maximum number of best fits in a time window was selected and indicated with a * in the figure. All the responses explained by the selected variable were removed from the pool and the procedure was repeated on the residual data set. Selected variables whose marginal explanatory power was <5% were eliminated (Methods) and indicated with a ● in the figure. In the first four iterations, the procedure selected variables chosen juice, chosen value, offer A ORF$_{uniform}$ and offer B ORF$_{uniform}$, and no other variables were selected in subsequent iterations. **b** Percent of explained responses. The y-axis represents the percentage of responses explained at the end of each iteration. The total number of task-related responses (1378) corresponds to 100%. The number of responses explained by at least one of the variables included in the analysis (1245/1378=90%) is indicated with a dotted line

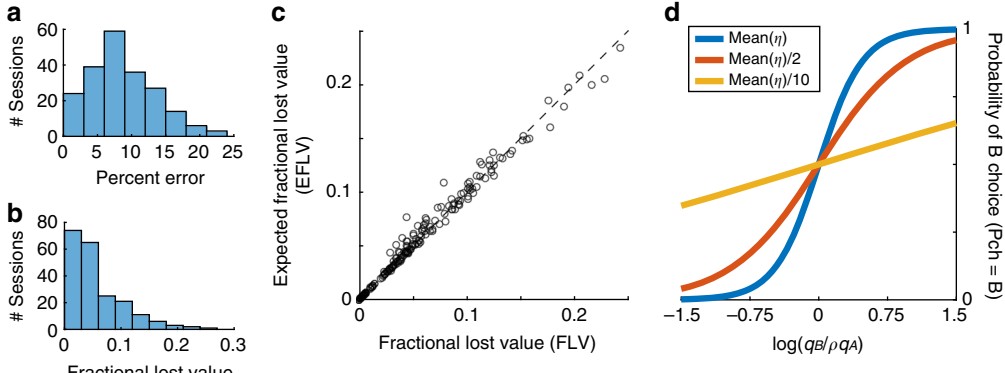

**Fig. 10** Analysis of fractional lost value. **a** Distribution of percent errors across sessions. On average across sessions, mean(percent errors)=8.7%. **b** Distribution of fractional lost value across sessions. On average across sessions, mean(FLV) = 0.054. **c** Expected vs. actual fractional lost value. Each data point represents one session. The expected fractional lost value (EFLV; y-axis) was almost identical to the fractional lost value (FLV; x-axis). On average across sessions, mean(EFLV) = 0.055. **d** Effects of decreasing the steepness of the sigmoid. The range on the x-axis is realistic for our experiments. The blue line is obtained with the mean sigmoid steepness measured in Exp.1 (mean($\eta$)). The red and yellow lines were obtained by dividing mean($\eta$) by 2 and 10, respectively

Institutional Animal Care and Use Committees at Harvard University (Exp.1) and Washington University in St Louis (Exp.2). No subject randomization or blinding during data analysis was used.

The procedures for Exp.1 have been described previously[4]. Briefly, one male (V, 9.5 kg) and one female (L, 6.5 kg) rhesus monkey participated in the experiment. Animals sat in an electrically insulated enclosure with the head restrained, and a computer monitor was placed in front of them at 57 cm distance. In each session, the monkey chose between two juices, labeled A and B, with A preferred. At the beginning of each trial, the animal fixated a center position on the monitor (Fig. 1a). After 0.5 s, two sets of colored squares appeared on the two sides of the center fixation. The two sets of squares represented the two offers, with the color associated with a particular juice type and the number of squares indicating the juice quantity. The animal maintained center fixation for a randomly variable delay (1–2 s), at the end of which the center fixation point was extinguished (go signal). The animal revealed its choice by making a saccade towards one of two targets located by the offers, and maintained peripheral fixation for an additional 0.75 s before the chosen juice was delivered. While animals performed in the task, we recorded the activity of individual neurons from the central OFC (see below). The range of quantities offered for each juice remained fixed within a session, while the quantity offered on any given trial varied pseudo-randomly. Across sessions, we used various juice pairs and various quantity ranges for the two juices. The minimum quantity was always zero drops (forced choice for the other juice), while the maximum quantity varied from session to session between 2 and 10 drops.

In Exp.2, animals performed essentially the same task, except that sessions were divided into two blocks of trials. One male (B, 9.0 kg) and one female (L, 6.5 kg) rhesus monkey participated in the experiment. The task was controlled through custom-written software based on Matlab (MathWorks)[52, 53] and gaze direction was monitored with an infrared video camera (Eyelink, SR research). The trial structure was the same as Exp.1, except that the initial fixation lasted 1.5 s. Each session included two trial blocks. The minimum offered quantity for each juice was always set to zero (forced choice for the other juice). The maximum quantity (and thus the range) varied from session to session and from block to block. In the second block, we either halved or doubled the range for one juice (A or B) while keeping the other range unchanged. This procedure resulted in a 2×2 design. Each block included 110–260 trials. In each block, an "offer type" was defined by a pair of offers (e.g., [1A:3B]); a "trial type" was defined by an offer type and a choice (e.g., [1A:3B, B]). The relative value of the two juices was computed from the indifference point (see below).

In principle, changes in relative value could arise from factors other than the value range. Exp.2 was designed to minimize three potential sources of choice bias. First, in previous work, we often noted that the relative value of any two juices tends to increase over the course of each day, presumably because animals become less thirsty. To deconfound changes in relative value due to changes in value range from this effect, we alternated sessions in which we increased or decreased the range of either juice A or juice B. The number of sessions for each of the 4 possible combinations was not-predetermined with a statistical method but was comparable ($\Delta A \rightarrow 2\Delta A$, 61 sessions; $\Delta B \rightarrow 2\Delta B$, 62 sessions; $2\Delta A \rightarrow \Delta A$, 49 sessions; $2\Delta B \rightarrow \Delta B$, 48 sessions). Second, within each trial block, monkeys might experience juice-specific satiety or diminishing marginal returns. Thus to isolate the behavioral effects of manipulating the value range, we ensured that in both trial blocks the animal drank the same relative amounts of the two juices. For example, if the animal drank juice A and juice B in quantity ratio 3:2 in the first block, we kept the same ratio 3:2 in the second block (see below). Third, we previously found that, all other things equal, monkeys tend to choose on any given trial the same juice they

chose in the previous trial (choice hysteresis)[5]. If the relative number of trials in which the animal chooses a particular juice varies from one block to the other, choice hysteresis could introduce a systematic bias. To avoid this confound, we ensured that the relative number of choices was the same in the two trial blocks.

The relative number of choices and the relative amount drunk by the animal for each juice were controlled by adjusting the frequency with which each offer type was presented. Specifically, offers were presented pseudo-randomly in mini-blocks of 20–30 trials. To fine-tune the balance between juice A and B, we kept track of the monkey's choices online. If the choice ratio or the relative amount of juice changed in the second block, the imbalance was corrected by adding forced choices of one of the two juices.

**Analysis of behavioral data**. Monkeys' choices generally presented a quality-quantity trade-off. If the two juices were offered in equal amounts, the animal would generally choose juice A (by definition). However, if sufficiently large quantities of juice B were offered against one drop of juice A, the animal would choose juice B. Choices were analyzed separately in each session (Exp.1) or in each trial block (Exp.2). The "choice pattern" was defined as the percentage of trials in which the animal chose juice B as a function of the log quantity ratio $\log(q_B/q_A)$, where $q_A$ and $q_B$ indicate the quantities of juices A and B. Each choice pattern was fitted with a sigmoid function:

$$P_{ch=B} = \int_{-\infty}^{X} N(0,1)\,dt$$
$$X = a_0 + a_1\log(q_B/q_A) \tag{5}$$

where $P_{ch=B}$ is the probability of choosing juice B and $N(0,1)$ is the standard normal distribution. The fit was done with Matlab function glmfit and link=probit. From the fitted parameters $a_0$ and $a_1$ we defined the relative value $\rho$ and the steepness of the sigmoid $\eta$ as follows:

$$\rho = \exp(-a_0/a_1)$$
$$\eta = a_1 \tag{6}$$

Given a set of offers, the expected payoff is directly related to $\eta$. In some simulations (Fig. 10), we reduced the sigmoid steepness (e.g., $a_1 \rightarrow a_1/10$) while keeping the relative value constant ($a_0/a_1 \rightarrow a_0/a_1$).

Exp.1 included 208 sessions. However, in some cases the choice patterns were saturated (i.e., the animal did not split decisions for any offer type, a situation referred to as "perfect separation"). In these cases, the sigmoid fit did not provide a reliable measure for $\eta$. Thus the analysis shown in Fig. 5 and Fig. 6 included only sessions for which choice patterns were not saturated (164 sessions).

In Exp.1, the minimum quantity offered for each juice was always 0, and we indicate maximum quantities with $Q_A$ and $Q_B$. We usually set $Q_A$ and $Q_B$ to approximately satisfy $\rho Q_A = Q_B$ (symmetric condition). However, this relation did not hold strictly, partly because the relative value $\rho$ was determined by the animal and fluctuated from session to session. Thus to test a theoretical prediction on choice variability and value range, we computed the geometric mean value range $\Delta \equiv (\rho\, Q_A\, Q_B)^{1/2}$ and we examined the relation between $\eta$ and $\Delta$. Since errors of measure affected both measures, standard regressions could not be used. We thus used Deming's regressions[54]. Variance ratios $\lambda$ were computed through error

propagation as follows:

$$\lambda = \frac{\mathrm{var}(\eta)}{\mathrm{var}\left((\rho Q_A Q_B)^{1/2}\right)} = \frac{(\delta\eta)^2}{Q_A Q_B (\delta(\rho^{1/2}))^2} = \frac{4a_1^4(\delta a_1)^2}{\rho Q_A Q_B (a_0 \delta a_1 - a_1 \delta a_0)^2} \quad (7)$$

where $\delta\rho$, $\delta\eta$, $\delta a_0$, and $\delta a_1$ are errors on the respective measures, and $\delta a_0$ and $\delta a_1$ are obtained as standard errors from the logistic regressions.

The relation between $\eta$ and $\Delta$ was also analyzed using alternative definitions for $\Delta$ including the simple mean $\Delta \equiv (\rho Q_A + Q_B)/2$ and the log geometric mean $\Delta \equiv \log (\rho Q_A Q_B)/2$, adjusting variance ratios accordingly. All variants of the analysis provided very similar results.

One concern was whether the relation between choice variability and value range (Fig. 6) was direct or reflected some other dependency. We considered two issues. First, Fig. 6 includes sessions with different juice pairs, with different typical values of $\rho$. In principle, choice variability could vary from juice pair to juice pair in a way that induces the relation observed in Fig. 6. Second, for any given juice pair, value range ($\Delta$), relative value ($\rho$) and sigmoid steepness ($\eta$) are all inter-related by definition (Eq. 6) and because value ranges were often chosen non independently of $\rho$ (in many sessions we set $Q_A = 3$ and chose $Q_B$ such that $Q_B \approx \rho\, Q_A$). Thus the relation between $\eta$ and $\Delta$ (Fig. 6) might simply reflect fluctuations in $\rho$. To address these concerns, we divided sessions in different sets based on the animal and on the juice pair. We considered only sets with $\geq 5$ sessions, and our data included 12 such sets (6 from each monkey). We then analyzed each set of sessions separately. First, we verified that the relation between $\Delta$ and $\eta$ held true for individual juice pairs (Supplementary Fig. 5). Second, to assess whether this relation simply reflected fluctuations in $\rho$, we used multilinear regression. For each set, we regressed $\eta$ on $\rho$ and then on $\Delta$ in a stepwise way. The coefficient obtained from the second regression ($\beta_\eta$) essentially quantified the correlation between $\eta$ and $\Delta$ not explained by fluctuations of $\rho$.

**Analysis of neuronal data.** Neuronal data were collected in Exp.1. The data set included 931 cells from central OFC (area 13). The number of cells recorded was not pre-determined using statistical methods. Firing rates were analyzed in seven time windows aligned with different behavioral events (offer, go, juice). A "neuronal response" was defined as the activity of one cell in one time window as a function of the trial type. Neuronal responses were computed by averaging firing rates across trials for each trial type.

In a previous study[4], we conducted a series of analyses to identify the variables encoded in this area. First, we submitted each neuronal response to a 3-way ANOVA (factors: position x movement direction x offer type), and we imposed a significance threshold $p < 0.001$. In total, 505 (54%) neurons passed the criterion for factor offer type in at least one time window. Pooling neurons and time windows, 1379 neuronal responses passed the ANOVA criterion, and subsequent analyses were restricted to this data set. Second, we defined a large number of variables including variables related to individual juices (offer value A, offer value B, chosen value A, chosen value B), other value-related variables (chosen value, other value, value difference, value ratio, total value), number-related variables (chosen number, other number, and so on) and choice-related variables (chosen juice). We performed a linear regression of each response onto each variable. If the regression slope differed significantly from zero ($p < 0.05$), the variable was said to "explain" the response. Because variables were often correlated with each other, the same neuronal response was often explained by more than one variable. Thus for each response we also identified the variable that provided the best fit (i.e., the highest $R^2$). Third, we proceeded with a variable selection analysis to identify a small subset of variables that best explained the neuronal population. We adapted two methods originally developed for multi-linear regressions in the presence of multi-collinearity, namely stepwise and best-subset[55, 56]. In the stepwise method, we identified the variable and time window that provided the highest number of best fits, and removed from the data set all the responses explained by that variable. We then repeated the procedure until when the number of responses explained by additional variables was < 5%. While intuitive, the stepwise procedure did not guarantee optimality. In contrast, the best-subset procedure (an exhaustive procedure) guaranteed optimality. In this case, for $n = 1, 2, 3, \ldots$ we computed the number of responses and the total $R^2$ explained by each subset of n variables. The best subset was identified as that which explained the highest number of responses or the maximum total $R^2$. In the original study, the stepwise and best-subset procedures identified the same 4 variables, namely offer value A, offer value B, chosen value and chosen juice. Fourth, we conducted a post-hoc analysis. While the explanatory power of variables included in the best subset was (by definition) higher than that of any other subset of variables, the procedure did not guarantee that this inequality was statistically significant. The post-hoc analysis addressed this issue by comparing the marginal explanatory power of each variable in the best subset with that of other, non-selected variables (binomial test). In the original study and subsequent work[32, 57, 58], we found that the explanatory power of offer value A, offer value B, chosen value and chosen juice was statistically higher than that of any other variable.

The results of these analyses provided a classification for neuronal responses. Specifically, each neuronal response was assigned to the variable that explained the response and provided the highest $R^2$. Thus we identified 447 offer value, 370

chosen value, and 268 chosen juice responses. Subsequent analyses of the same data set demonstrated range adaptation[6] and quantified noise correlations[30].

Previous work suggested that the encoding of value in OFC was close to linear[4, 6]. In this study, we conducted more detailed analyses to quantify how neuronal responses departed from linearity. Furthermore, we compared the curvature measured in neuronal responses with that measured for the cumulative distribution of the encoded values. The cumulative distribution of encoded values was taken as a benchmark by analogy with sensory systems[14, 19]. For each offer value response, we identified the value levels present in the session (i.e., the unique values). We then calculated the corresponding number of trials, divided it by the total number of trials in the session, and computed the corresponding cumulative distribution function (ntrials$_{CDF}$). For each value level, we also averaged the firing rates obtained across trials. The range of offered values and the range of firing rates varied considerably from session to session and across the population (Supplementary Fig. 3). Thus to compare the results obtained for different responses, we normalized offer value levels, neuronal firing rates, and ntrials$_{CDF}$. Value levels were simply divided by the maximum value present in the session. For example, normalized values for juice B were defined as $q_B/Q_B$. For neuronal firing rates, we performed the linear regression $y = a_0 + a_1\, x$, where $x$ are normalized value levels. Firing rates were thus normalized with the transformation fr $\rightarrow$ (fr$-a_0$)/$a_1$. Similarly, for ntrials$_{CDF}$, we performed the linear regression $y = b_0 + b_1\, x$, and we normalized them with the transformation ntrials$_{CDF} \rightarrow$ (ntrials$_{CDF}-b_0$)/$b_1$. Examples of normalized firing rates and normalized ntrials$_{CDF}$ are illustrated in Fig. 1d, g. To estimate the overall curvature, we fit each normalized response function with a 2D polynomial and compared the quadratic coefficient ($\beta_2$) with that obtained from fitting the corresponding normalized ntrials$_{CDF}$. To estimate the overall S shape, we fit the normalized response function with a 3D polynomial and compared the cubic coefficient ($\beta_3$) with that obtained for the corresponding normalized ntrials$_{CDF}$. These measures were used for the population of offer value responses (Fig. 2). Separately, we repeated these analyses for the population of chosen value responses (Supplementary Fig. 2).

Theoretical considerations indicated that optimal response functions in our experiments would have been step functions (Fig. 7b, c), contrary to our observations. One concern was whether empirical response functions were optimal on average across sessions, if not for any particular session. Notably, the relative value $\rho$ varied from session to session, largely because we used a variety of different juice pairs. Recent work indicates that the same neurons are associated with different juices in different sessions, with remapping dictated by the preference ranking[58]. In any given session, the optimal offer value B response function would have been a step function with step at $x = \rho$. However, since $\rho$ varied from session to session, the resulting optimal response function would have been more gradually increasing. In fact, if the distribution of $\rho/Q_B$ across session had been uniform in the interval [0 1], the mean optimal response for offer value B neurons would have been linear. An important caveat is that the rationale that would justify linear offer value B responses did not hold for offer value A responses. In any case, we examined the distribution of $\rho/Q_B$ (Fig. 7d). For offer value B, the mean optimal response function was computed as the cumulative distribution function for $\rho/Q_B$.

Importantly, the data sets included in Fig. 2 and Supplementary Fig. 3 were originally selected based on a procedures that only considered linear encoding of value[4]. To assess the functional form of neuronal responses without bias in favor of linearity, we repeated the variable selection procedures described above including in the analyses all the variables defined in this study. The variable selection analysis was still based on linear regressions. However, firing rates were regressed on several non-linear value variables (response functions). The analysis included linear response functions (offer A, offer B), cumulative distribution functions of offer values (offer A ntrial$_{CDF}$, offer B ntrial$_{CDF}$), variance-equalized versions of the cumulative distribution functions of offer values (offer A ntrial$_{CDF}$ VE, offer B ntrial$_{CDF}$ VE; see below), optimal response functions (offer A ORF, offer B ORF), mean optimal response functions across sessions (offer A mean(ORF), offer B mean(ORF)), and optimal response functions obtained under the assumption that the joint distribution of offer values is uniform and symmetric (offer A ORF$_{uniform}$, offer B ORF$_{uniform}$; see below). In addition, the analysis included chosen value, the cumulative distribution function for chosen values (chosen value ntrial$_{CDF}$), a variance-equalized version of the cumulative distribution function for chosen values (chosen value ntrial$_{CDF}$ VE), and variables other value, value difference, value ratio, total value and chosen juice (20 variables total).

The last part of Supplementary Note generalizes the theory of optimal coding by relaxing the assumption of linear response functions. In principle, this allows one to compute the optimal response functions for any joint distribution of offers. Optimal response functions for the symmetric uniform distribution (ORF$_{uniform}$) were computed numerically in Matlab.

We restricted the variable selection analysis to responses that passed the ANOVA criterion (N=1379, see above) and we regressed each neuronal response on each variable. For variance-equalized variables, we first computed the square root of the firing rates and then performed the linear regressions[59, 60]. For variable selection we used the two procedures described above, and we refer to previous work for additional details[4]. The best-subset method can be based either on the number of responses or on the total $R^2$ explained by each subset. In previous studies, the two metrics provided similar results. Here we found that the results obtained based on the total $R^2$ were more robust, probably because the present

analysis aimed at providing a better fit for the neuronal responses as opposed to explaining more responses. The best-subset procedures and post-hoc analyses were performed on collapsed variables[4]. The variable selection analyses were conducted twice. First, we included all the variables described above except those based on $ORF_{uniform}$. In this case, linear response functions performed significantly better than all the other variables (Supplementary Table 1). Second, we added in the analysis the variables based on $ORF_{uniform}$. In this case, the performance of $ORF_{uniform}$ variables was better than, but statistically indistinguishable from that of linear variables. It was significantly better than that of all the other variables (Supplementary Table 2). Both analyses are described in the Results. Figures 8 and 9 refer to the analysis that included all 20 variables.

**Code availability**. The code used for data analysis and simulations is available from the corresponding author upon reasonable request.

**Data availability**. The data that support the findings of this study are available from the corresponding author upon reasonable request.

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

## Acknowledgements

We thank Heide Schoknecht for help with animal training, Nicolas Brunel and Daniel Chicharro for helpful conversations, and Harold Burton, Arno Onken, and Stefano Panzeri for comments on earlier versions of the manuscript. This work was supported by the National Institutes of Health (grant numbers R01-DA032758 and R01-MH104494 to C.P.-S. and grant numbers T32-GM008151 and F31-MH107111 to K.E.C.) and by the National Science Foundation (grant SES-1357877 to A.R.). This work was partly conducted while C.P.-S. was a visiting fellow at the Italian Institute of Technology.

## Author contributions

A.R. and C.P.-S. designed the study; K.E.C, X.C., and C.P.-S. collected the experimental data; K.E.C. and C.P.-S. analyzed the experimental data; A.R. and C.P.-S. developed the mathematical formalism; A.R. and C. P.-S. wrote the manuscript.

## Additional information

**Competing interests:** The authors declare no competing financial interest.

