## [Peer Review File · Nature Communications]

Reviewers' comments:

Reviewer #1 (Remarks to the Author):

Rustichini A et al., Neuronal adaptation and optimal coding in economic decisions

The authors present a detailed study of the tuning properties of OFC offer value neurons and their relationship to optimal encoding for decisions. They show that OFC neurons encode offer values approximately linearly, and that they rescale according to the range of values. The rescaling does not lead to biased decision making, and it improves decision accuracy.

This paper address several important issues and does so within a well-developed, straightforward and clearly presented analytical framework. I have no overall concerns with the study, although I do have a number of specific comments that may help clarify the results and interpretation.

Comments

1. It might be worth laying out the definition of optimality upfront, descriptively. It's given in the equations. Optimality is not conveying the most information about the offers. In this sense the tuning functions of the neurons are not quite consistent with the CDF of the offer distribution. Optimality is defined as the animals consistently choosing their most preferred option. In this sense, the optimal tuning functions just encode the animal's choices in a binary fashion in several of the experimental cases considered (Fig. 6a-c). This is briefly addressed in the paragraph on page 7 where the sentence, "In retrospect, this finding..." Maybe just bring this to the front and use it to frame the approach in general.

2. To some extent, this optimality argument feels a bit circular, since it's not defined relative to an external variable. Rather it's defined relative to the subjective preferences of the animal. If we believe the brain is controlling behavior, then the activity (perhaps in OFC offer value cells, but maybe in combination with some other activity somewhere) must be sufficient to drive the animal's choices. If neurons were "more optimal" then a smaller population could drive the animal's choices? Or perhaps the animal's choice functions would have steeper slopes?

3. The curve fits in Fig. 1 c, d, f, g and the population fits reported in Fig. 2 only have a few degrees of freedom. In some cases only 4. I realize that many trials go into each point, but those are not true degrees of freedom of the actual function that generates the data, and therefore the fit of the function to the data is also limited. This is a picky point, but were there any differences in the fits of the 2nd and 3rd order terms when there were more degrees of freedom (for example, in 1g compared to 1d)? If one only has 4 degrees of freedom, and a linear fit has 2, the higher order terms are unlikely to contribute much.

4. I wasn't sure why the indices in equation 1 were superscript on X and subscript on K? For consistency maybe use super for all?

5. For the analysis that shows that the offer value cells, "...did not adapt to maximize the payoff in each session." How much was actually lost? How much better, in terms of milliliters/trial or some related metric, would an ideal population have done? Again, maybe this is not a well posed question because it would depend on the population size.

Supplemental methods

These seem like they should be published separately in a more technical journal with referees that

would correspondingly be more appropriate to this material.

6. "This a setup..." -> "This setup..."

7. Just under equation 7, in Assumption 2.2, $X_A = X_A \text{ def } X$ should be $X_A = X_B \text{ def } X$.

8. Page 6. "We clarify now of the details..." -> "We clarify now the details..."

Reviewer #2 (Remarks to the Author):

This manuscript by Rustichini and colleagues examines the nature of adaptation in value coding of monkey OFC neurons and the relationship between this adaptation and optimal coding for choice behavior. This work follows a body of work from Padoa-Schioppa's group examining linear value coding and range adaptation in the orbitofrontal cortex. The key question this paper addresses is the potential optimality/purpose of value range adaptation: in sensory coding, adaptation that approximates the cumulative distribution of sensory variables is optimal in terms of efficiently encoding the available information; however, decision systems must implement choice and optimality can be thought of in terms of decision outcomes rather than coding alone.

Here, the authors report a series of theoretical and empirical results:

(1) Reanalyzing previously reported data, the authors show that offer value neurons show quasi-linear responses despite the experimental distribution of rewards being typically highly non-uniform.

(2) Using new experimental data, the authors show that despite linear range adaptation - which can introduce a choice bias as the coding one of two potential rewards changes with the range of presented offers - monkey choice behavior exhibits stable relative reward preferences. From these results, the authors conclude that the effects of range adaptation are corrected in the decision circuit.

(3) They present a theoretical argument - primarily in the Supplementary Material - that assuming linear response functions range adaptation maximizes expected payoff, even if the subsequent circuit rescales to remove adaptation-induced choice biases.

(4) They show that - in monkey behavior - received payoff is inversely related to the range of presented rewards.

(5) Finally, they show that the observed quasi-linear response functions do not resemble the step-function ORFs predicted for offer value cells, under the assumption of coding the joint distribution of rewards. Instead, they suggest that the observed response functions most closely resemble that predicted for symmetric uniform rewards.

These studies address a question of much interest: what is the relationship between PFC value coding and choice behavior, and can the form of this coding be understood in terms of efficient coding? The authors raise a salient and important point: that sensory systems are optimized for efficiency given stimulus statistics, but value coding neurons may be optimized with different constraints (payoff, or joint reward statistics). I think this paper has promise, but there are a few issues in the current version that need to be addressed. The two biggest issues are the assumption that OFC value coding neurons are involved in the decision process (which is the basis for concluding that there is corrective rescaling downstream in the circuit), and the validity of examining payoff-range relationships across sessions with different goods. Additionally, there are several areas where details can be cleared up by the authors.

Major points

(1) One of the major points of the paper is that OFC neurons exhibit range adaptation, but that monkey choice behavior does not exhibit the resulting, problematic choice biases that arise from the non-stationary value coding implied by range adaptation. The authors conclude that the decision circuit compensates for the effects of OFC range adaptation, perhaps by adjusting downstream synaptic efficacies (“We also showed that range adaptation is corrected within the decision circuit to avoid arbitrary choice biases.”).

However, there is a rather large caveat to this conclusion: it assumes that OFC neurons exhibiting range adaptation are causally involved in the choice process, but as far as I know this has yet to be demonstrated. One could argue that an equally plausible possibility is that 1) OFC neurons are range adapting but *not* involved in choice, but 2) the value coding and decision circuit neurons are not adapting and thus do not fall prey to arbitrary choice biases.

The data from Experiment 2 are still interesting, since they reaffirm the notion that the relative preference of goods is not dependent on the ranges of offered rewards. However, without causal evidence that OFC neurons drive choice, the link between range adaptation and behavior is unknown and the suggestion that correction occurs in the decision circuit is unfounded. I think the authors could argue that the relationship between value range and payoff (Fig. 5) suggests that range adaptation is part of the decision process, but see point(2) below about problems with that data interpretation.

(2) Perhaps the most interesting and potentially important data from the paper is the finding that monkey choice behavior is value range-sensitive. Past papers have documented range adaptation in OFC neurons but no clear corresponding behavioral correlates (Padoa-Schioppa 2009; Kobayashi et al 2010). However, there is a critical point of analysis that confounds the argument that payoff (or the inverse of sigma) is related to the value range.

Figure 5 and the accompanying analyses aim to compare payoffs (or equivalently choice curve slopes, or sigma) across different sessions with different value ranges. However, in the data (from Padoa-Schioppa and Assad, 2006), there were ~12 juice types and individual sessions presented single pairs of these types. The fundamental issue: how can the behavior in different sessions be compared when different juices are used across sessions? Specifically, how are the x axes in Fig. 5 co-aligned for different sessions? Similarly, how can the sigmoid fits be compared across different sessions with different juice types?

Both Fig. 5 and the sigmoid fits are performed with each session analyzed in objective units of q_B (drops of juice B). But this assumes that a given change in juice B in one session is equivalent to a given change in juice B in a different session, in *subjective* terms. However, there is no reason to think that the subjective change in value of a given quantity change in grape juice is equivalent to that of the same objective quantity change in peppermint tea. One could argue that utilizing ratios of goods (in equivalent B units) as the authors do could alleviate the problem, but this approach assumes that the linear scaling of utility for separate goods are the same. If the x axes in Fig. 5a,b cannot be aligned across sessions, then it is impossible to compare payoff/choice stochasticity across different sessions.

On a related note, I’m not sure that value ranges can be compared across sessions either. For example the correlational analyses in Fig. 5c,d rely on a measure of mean value range, defined as $(\rho * Q_A + Q_B) / 2$. This means that each session range is defined in objective units (drops) of a particular good (good B). I don’t see how ranges across sessions can be compared when they are in different objective (different goods) and subjective (the utility of those goods, which are not compared) units.

Given these caveats, the data from Expt. 2 (Fig. S5) *can* be analyzed because they address offer range changes without changing goods. The Supplementary Material provides some evidence that there is such a relationship in a limited set of the data. However, to make the argument that there is a relationship between payoff and value range, the authors would have to do a fuller analysis of all the data from the experiment (i.e. including the sessions where A changes).

(3) The discussion raises several issues that are currently unaddressed, and the authors should clear up for the reader. First, it's not clear to me which optimization goal is being served by range adaptation: the paper argues that decision model maximizes payoff at different value ranges, but also presents results that suggest that quasi-linearity mimics ORF for joint uniform reward distributions. In the former, optimization is for decision outcomes, whereas in the latter, optimization is - as for sensory systems - for input (reward) coding.

Second, the relationship between joint reward distribution-based response functions and menu independence (Discussion, pg 8-9) is puzzling to me. The authors note that OFC value coding is menu invariant, and that each neuron adapts to its own value range, suggesting that "neurons associated with one particular good are blind to every aspect of the other good". However, an ORF that is tuned to the *joint* uniform distribution of rewards must take into account the range of the other good. This seems like a contradiction to me, perhaps the authors can clear this up?

Other points

(1) The abstract stated that "we show that neuronal responses are quasi-linear even when optimal tuning functions would be highly non-linear.". It's crucial here that the authors state what is meant by optimal: is it optimal in the classic sensory CDF sense, or optimal in terms of expected choice payoff?

(2) Clarity of the model and theoretical predictions. In the synopsis of model results (pg 5-6), the main point is that maximal expected payoff occurs when there is complete range adaptation. In other words, given a range of offer values, maximum payoff occurs with complete adaptation of firing rates. However, the prediction in the next section that payoff and value range are inversely related is not addressed and it is not clear from where it arises.

(3) How does a nonlinear value function of quantity (drops) affect these interpretations? One of the points of the paper is that there is a functional rigidity to OFC responses: linear coding of offered rewards (in drops). However, much work in the economics literature suggests that people and animals have nonlinear utility functions. One wonders if the linearity in response to rewards here is an experimental byproduct of the quantization of juice delivery in drops; if juices were delivered as different volumes rather than different numbers of drops, perhaps value coding would be nonlinear? I certainly don't expect the authors to provide different experimental data, but could they address the relationship between linear value coding and nonlinear utility functions?

(4) How ρ (relative value) is defined is not clear from the text. I understand that it is quantified as the slope of the indifference line (Fig. 4a) between offer A and offer B, and essentially measures the tradeoff between objective amounts of A and B, but it should be clarified in the text when first discussed (pg 5?).

(5) Can the authors clarify what they mean by "payoff" in the main text? It is never clearly defined, but presumably this is a measure of how often the animal chooses the higher subjectively valuable reward (which itself is defined by the animal's behavior in the block).

(6) Figure 5. Can the authors clarify what the blue-green shading depicts? Does it represent the size

of the mean value range? Please either state explicitly in the legend, or provide a visual colorbar.

(7) Figure 6e. The legend describes the case where $\rho=2$, but the dashed line in panel (e) depicts $\rho=1$.

(8) Optimal response functions (Fig 6).

- Not sure how they are derived. The idea that sensory neural tuning curves approximate the CDF of the encoded statistic makes specific certain assumptions, such as the desire for a uniform utilization of output FRs (maximizing entropy/information). Can the authors be clearer about how the ORFs are derived?

(9) In the presentation summarizing the model implications (pg 6), the authors discuss symmetric and asymmetric cases, but it would help the reader if the explicit difference between the two are stated in plain terms. I believe symmetric describes instances when the range of the two goods are equivalent (in good B-equivalent units), and asymmetric describes when they are not.

Reviewer #3 (Remarks to the Author):

In this manuscript, Rustichini et al. present deeper investigations of data and results related to a series of publications from the Padoa-Schioppa lab, in which orbitofrontal (OFC) neurons in monkeys are found to encode decision parameters in choices involving quantity-quality trade-offs. The authors conducted a series of analyses collectively aimed at elucidating the nature of value coding in OFC neurons, particularly in the context of range adaptation.

The previous studies from this group report three coding schemes common among OFC neurons (offer value, chosen value, chosen juice), and the present manuscript focuses on those encoding 'offer value', referring to the value of one of two juice rewards available in a given task session. These neurons are believed to encode pre-decision variables, such that a choice can be made by comparing the relative firing rates of two 'offer value' populations coding for each option. It has previously been reported that they exhibit range adaptation. That is, their firing rates do not faithfully reflect the absolute quantity of juice, but rather adapt so that baseline and maximal firing rates roughly map onto the minimum and maximum quantities available in a given session, with intermediate values linearly related.

Given this background, the authors first show that firing rates of offer value neurons adapt based on the range of values available in a session, but not the frequency with which these values are encountered (i.e. their firing rates are "quasi-linear" with respect to value). This result is novel if not entirely surprising. Indeed, it could provide interesting support for recent ideas that OFC neurons form "cognitive state space maps", in that the neurons appear to respond based on stored knowledge of the range of possible values (i.e. 'maps') rather than being influenced by the statistics of individual trials.

This result is loosely tied to the final part of the paper, in which the authors reanalyzed their data to assess non-linear explanations of firing rates, none of which outperformed the linear models. Only explanatory variables based on uniform joint distributions of the two offered values performed comparably to the linear models. However, whether a joint distribution of offer values provides a better description of neuron firing is not precisely resolved here, as both models fit the neural data well and were not statistically distinguishable.

A second theme of the paper was investigating an apparent paradox of adaptive coding among these

neurons, which can be paraphrased as follows: if offer value neurons adapt their firing rates based on the available value range of one juice, and if we assume that the choice is a result of a comparison of offer value A vs. offer value B firing rates, then the indifference point in choices would shift depending on the range of values available. Because this shift is not observed behaviorally, it might be concluded that the choice is not strictly based on the relative firing rates of offer value neurons. This is an interesting observation. To drive this home, the authors tested monkeys in a nearly identical task, in which daily sessions were blocked so that the range of values in block 2 either increased or decreased. They showed that, behaviorally, the indifference point did not shift with different value ranges, implying there must be a correction somewhere in the decision circuitry. The authors hypothesize a downstream correction mechanism. While this explanation is speculative and not entirely satisfying, I think it is beyond the scope of this paper to find and demonstrate such a corrective mechanism.

The final piece of the story was a demonstration that range adaptation maximized expected payout. This argument was difficult to follow, in part because much of it was relegated to supplementary material, and also because it was not immediately clear how the authors defined and quantified 'payoff'. I gather that it's based on the value obtained as an abstract quantity not tied to the actual juice (i.e. 'chosen value?'), but this isn't really spelled out anywhere. If this interpretation is correct, shouldn't it be closely tied to the maximum value available in a session, not just the value range?

Further comments

(1) Overall I found this a challenging paper. There are interesting investigations here, but the manuscript as written relies heavily on self-referential ideas and terms making it difficult for the reader to parse what exactly was hypothesized and how it was addressed, even more so if they do not specialize in the area. It would be significantly improved by making it more accessible to a general audience.

(2) The authors seem to expect that these neurons should behave in a manner that is optimized for performing binary choices, for example in assuming that the optimal coding strategy is a step function that relates to the value of one juice option, with the step at the indifference point. This would, indeed, be optimal in the case of a familiar 2-option choice. But in how many other situations would this be optimal? It seems that having more parametric information, as provided by a response that varies more linearly with value (as was actually observed in the neurons), would be optimal for a wider range of behaviors. So in effect, is the argument that the neurons "should" respond via a step function something of a straw man?

Introductory remarks on this revision

We sincerely thank the three Reviewers for their time and for their thoughtful comments. Addressing them has significantly improved the manuscript and we are grateful for it.

One recurrent comment from the Reviewers was that we did not introduce and clearly spell out the concepts of expected payoff and optimality, which are both central to this study. We completely agree with this critique, and we realize that this issue generated some confusion in the original ms. To address this point we revised several sections of the ms.

First, we added at the beginning of the Results the section *Relative value, choice variability and expected payoff*, where we provide both an intuition and a formal definition for the expected payoff. In a nutshell, the behavior in any given session is fitted with a sigmoid function. The flex of the sigmoid reveals the relative value (i.e., the quantity ρ of juice B that makes the animal indifferent between 1A and ρ B). The width of the sigmoid provides a measure of choice variability. If the sigmoid is steep (low choice variability), the animal always chooses the higher value; if the sigmoid is shallow (high choice variability), there is a significant fraction of trials in which the animal chooses the lower value. On any given trial, the payoff is equal to the chosen value. Thus given a set of offers and a sigmoid function, the expected payoff is equal to the chosen value averaged across trials. Importantly, the expected payoff is closely related to the sigmoid width: the wider the sigmoid, the lower the expected payoff. The two central paragraphs of this new section (p.4) are as follows:

Choice patterns often presented some variability. For example, consider in Fig. 1b offers 6B:1A. In most trials, the animal chose juice B, consistent with the fact that the value of 6B was higher than the value of 1A. However, in some trials, the animal chose the option with the lower value. Similarly, in some trials, the animal chose 3B over 1A. In each session, the width of the fitted sigmoid, referred to as σ , quantified choice variability (Methods).

In any given trial, we define the payoff as the value chosen by the animal. Thus given a set of offers and a sigmoid function, the expected payoff is equal to the chosen value averaged across trials. Importantly, the expected payoff is inversely related to choice variability, and thus inversely related to the sigmoid width. When the sigmoid is steeper, choice variability is lower, and the expected payoff is higher; when the sigmoid is shallower, choice variability is higher, and the expected payoff is lower. Notably, the relative value of two juices is entirely subjective. In contrast, a key aspect of the expected payoff is objective: Given a set of offers, a relative value and two sigmoid functions, the steeper sigmoid yields higher expected payoff.

Second, we revised the Introduction to define optimal coding for economic decisions. In a nutshell, neurons encoding the offer values are optimally tuned if they ensure maximal expected payoff. The key passage of the Introduction (p.3) is as follows:

Because they constitute the input layer of the decision circuit, [*offer value cells*] are in some ways analogous to sensory cells. However, the behavioral goal subserved by *offer value cells* differs from that subserved by neurons in sensory systems. For the purpose of accurate perception, sensory neurons are optimally tuned if they transmit maximal information about the

external world(Barlow, 1961; Laughlin, 1981; Simoncelli and Olshausen, 2001). This goal is achieved if tuning functions match the cumulative distribution of the encoded stimuli(Brenner et al., 2000; Laughlin, 1981). In contrast, the purpose of a subject performing economic decisions is to maximize the payoff (i.e., the chosen value). Thus *offer value* cells are optimally tuned if they ensure maximal expected payoff.

Other changes made to address the specific comments of the three Reviewers are detailed in the following pages.

Reviewer #1 (Remarks to the Author):

The authors present a detailed study of the tuning properties of OFC offer value neurons and their relationship to optimal encoding for decisions. They show that OFC neurons encode offer values approximately linearly, and that they rescale according to the range of values. The rescaling does not lead to biased decision making, and it improves decision accuracy.

This paper address several important issues and does so within a well-developed, straightforward and clearly presented analytical framework. I have no overall concerns with the study, although I do have a number of specific comments that may help clarify the results and interpretation.

We thank Reviewer 1 for the overall evaluation and for the specific comments.

Comments

1. It might be worth laying out the definition of optimality upfront, descriptively. It's given in the equations. Optimality is not conveying the most information about the offers. In this sense the tuning functions of the neurons are not quite consistent with the CDF of the offer distribution. Optimality is defined as the animals consistently choosing their most preferred option. In this sense, the optimal tuning functions just encode the animal's choices in a binary fashion in several of the experimental cases considered (Fig. 6a-c). This is briefly addressed in the paragraph on page 7 where the sentence, "In retrospect, this finding..." Maybe just bring this to the front and use it to frame the approach in general.

We completely agree with this comment and we extensively revised the ms accordingly. For details, please see the first two pages of this document (*Introductory remarks on this revision*).

2. To some extent, this optimality argument feels a bit circular, since it's not defined relative to an external variable. Rather it's defined relative to the subjective preferences of the animal. If we believe the brain is controlling behavior, then the activity (perhaps in OFC offer value cells, but maybe in combination with some other activity somewhere) must be sufficient to drive the animal's choices. If neurons were "more optimal" then a smaller population could drive the animal's choices? Or perhaps the animal's choice functions would have steeper slopes?

Optimality in economic decisions is essentially synonymous of consistency. If the animal is offered 1A:3B, there is no correct answer. However, if the animal is indifferent between 1A and 3B, it *should* choose 4B over 1A. Failure to do so is suboptimal. In other words, the behavior becomes closer to optimal as choice variability becomes closer to zero (i.e., the sigmoid becomes steeper).

Multiple factors contribute to choice variability (Padoa-Schioppa, 2013). The present study focuses on neuronal noise in the encoding of offer values. In the framework of a linear decision model and under reasonable assumptions, the theory presented in the Supplementary Material provides explicitly the slope of the sigmoid as a function of the value ranges, the maximum firing rate and the noise correlation between pairs of *offer value* cells (Eq.28). This equation contains the answer to the Reviewer's question. For given value ranges, neurons would be "more

optimal" if the maximum firing rate (v) was larger and/or if noise correlations ($\chi = \xi/4$) were lower. This is actually very intuitive: If neurons had a larger range of possible firing rates, or if they were less correlated with each other, their signal would be more reliable and decisions would improve. However, we emphasize that this statement and the whole theory takes into account (i.e., does not overlook) the challenges posed by adaptation.

All this assumes that the number of neurons is very large ($n \rightarrow \infty$). If the number of neurons was significantly smaller, the behavior would be less optimal. However, practically speaking, the limits imposed by noise correlation are more stringent than those imposed by the fact that the number of cells is not infinite (Shadlen et al., 1996). This is true even though noise correlation in OFC is significantly lower than that in area MT ($\xi \approx 0.01$ versus $\xi \approx 0.14$).

3. The curve fits in Fig. 1 c, d, f, g and the population fits reported in Fig. 2 only have a few degrees of freedom. In some cases only 4. I realize that many trials go into each point, but those are not true degrees of freedom of the actual function that generates the data, and therefore the fit of the function to the data is also limited. This is a picky point, but were there any differences in the fits of the 2nd and 3rd order terms when there were more degrees of freedom (for example, in 1g compared to 1d)? If one only has 4 degrees of freedom, and a linear fit has 2, the higher order terms are unlikely to contribute much.

The short answer is that the result illustrated in Fig.2 does not depend on the number of offer values. To elaborate, the number of values offered varied from cell to cell, as indeed noted by Reviewer 1 in Fig.1. In fact, the number of offer values was typically lower for juice A than for juice B. To address the Reviewer's question, we repeated the analysis of Fig.2 dividing responses in two groups, depending on whether the number of offer values was low (# offer values ≤ 5) or high (# offer values ≥ 6). Fig.S1gh shows the result of this analysis. Notably, departures from linearity were present in both groups, but they were more pronounced for the first group of responses (# offer values ≤ 5). The reason for this last observation is that low-value offers were predominant for both juices, but even more so for juice A than for juice B.

4. I wasn't sure why the indices in equation 1 were superscript on X and subscript on K? For consistency maybe use super for all?

No profound reason, just consistency with the Supplementary Material. The variable K in Eq.1 of the main text corresponds to variable J in Eq.1 of the Supplement. For J we use a subscript because we also need to write J^2 (Supplementary Material, Eq.3) and it makes the notation graphically easier. For this reason, we elected to leave the notation as in the original submission.

5. For the analysis that shows that the offer value cells, "...did not adapt to maximize the payoff in each session." How much was actually lost? How much better, in terms of milliliters/trial or some related metric, would an ideal population have done? Again, maybe this is not a well posed question because it would depend on the population size.

Very interesting question -- thanks! We addressed it based on the data and we included in the revised ms a new section that describes our results (see p.9-10, section *The cost of rigidity and the benefit of adaptation*).

In a nutshell, we defined a metrics named *fractional lost value* (FLV), which normally varies between 0 and 1. Specifically, $FLV = 0$ if the animal always chooses the higher value and $FLV = 1$ if the animal always chooses randomly. Thus FLV quantifies the fraction of value lost to choice variability. We find that the FLV due to suboptimal response functions was relatively small (≤ 0.05). In contrast, the FLV expected under incomplete range adaptation would be quite large. For example, if the inverse temperature (parameter a_I in Eq.4) was 10 times smaller than observed experimentally, $FLV = 0.52$. These results suggest that a quasi-linear, but range adapting encoding of offer values is sufficient to ensure close-to-optimal behavior.

Supplemental methods

These seem like they should be published separately in a more technical journal with referees that would correspondingly be more appropriate to this material.

We had considered turning the Supplementary Material into a separate paper before the original submission, and we gave more thought to this issue now. In the end, we think that the main text and the Supplementary Material are very intimately linked, and that having the former without the latter would impoverish this paper. One strength of this paper is that theory and data go hand in hand, and loosing it seems like a shame. Thus we decided to leave the Supplementary Material here.

6. "This a setup..." -> "This setup..."

Corrected

7. Just under equation 7, in Assumption 2.2, $X_A = X_A \text{ def } X$ should be $X_A = X_B \text{ def } X$.

Corrected

8. Page 6. "We clarify now of the details..." -> "We clarify now the details..."

Corrected

Reviewer #2 (Remarks to the Author):

This manuscript by Rustichini and colleagues examines the nature of adaptation in value coding of monkey OFC neurons and the relationship between this adaptation and optimal coding for choice behavior. This work follows a body of work from Padoa-Schioppa's group examining linear value coding and range adaptation in the orbitofrontal cortex. The key question this paper addresses is the potential optimality/purpose of value range adaptation: in sensory coding, adaptation that approximates the cumulative distribution of sensory variables is optimal in terms of efficiently encoding the available information; however, decision systems must implement choice and optimality can be thought of in terms of decision outcomes rather than coding alone.

Here, the authors report a series of theoretical and empirical results:

(1) Reanalyzing previously reported data, the authors show that offer value neurons show quasi-linear responses despite the experimental distribution of rewards being typically highly non-uniform.

(2) Using new experimental data, the authors show that despite linear range adaptation - which can introduce a choice bias as the coding one of two potential rewards changes with the range of presented offers - monkey choice behavior exhibits stable relative reward preferences. From these results, the authors conclude that the effects of range adaptation are corrected in the decision circuit.

(3) They present a theoretical argument - primarily in the Supplementary Material - that assuming linear response functions range adaptation maximizes expected payoff, even if the subsequent circuit rescales to remove adaptation-induced choice biases.

(4) They show that - in monkey behavior - received payoff is inversely related to the range of presented rewards.

(5) Finally, they show that the observed quasi-linear response functions do not resemble the step-function ORFs predicted for offer value cells, under the assumption of coding the joint distribution of rewards. Instead, they suggest that the observed response functions most closely resemble that predicted for symmetric uniform rewards.

These studies address a question of much interest: what is the relationship between PFC value coding and choice behavior, and can the form of this coding be understood in terms of efficient coding? The authors raise a salient and important point: that sensory systems are optimized for efficiency given stimulus statistics, but value coding neurons may be optimized with different constraints (payoff, or joint reward statistics). I think this paper has promise, but there are a few issues in the current version that need to be addressed. The two biggest issues are the assumption that OFC value coding neurons are involved in the decision process (which is the basis for concluding that there is corrective rescaling downstream in the circuit), and the validity of examining payoff-range relationships across sessions with different goods. Additionally, there are several areas where details can be cleared up by the authors.

We thank Reviewer 2 for the overall evaluation and for the specific comments.

Major points

(1) One of the major points of the paper is that OFC neurons exhibit range adaptation, but that monkey choice behavior does not exhibit the resulting, problematic choice biases that arise from the non-stationary value coding implied by range adaptation. The authors conclude that the decision circuit compensates for the effects of OFC range adaptation, perhaps by adjusting downstream synaptic efficacies (“We also showed that range adaptation is corrected within the decision circuit to avoid arbitrary choice biases.”).

However, there is a rather large caveat to this conclusion: it assumes that OFC neurons exhibiting range adaptation are causally involved in the choice process, but as far as I know this has yet to be demonstrated. One could argue that an equally plausible possibility is that 1) OFC neurons are range adapting but *not* involved in choice, but 2) the value coding and decision circuit neurons are not adapting and thus do not fall prey to arbitrary choice biases.

The data from Experiment 2 are still interesting, since they reaffirm the notion that the relative preference of goods is not dependent on the ranges of offered rewards. However, without causal evidence that OFC neurons drive choice, the link between range adaptation and behavior is unknown and the suggestion that correction occurs in the decision circuit is unfounded. I think the authors could argue that the relationship between value range and payoff (Fig. 5) suggests that range adaptation is part of the decision process, but see point(2) below about problems with that data interpretation.

Reviewer 2 makes a valid and important point. The gold standard for establishing a causal link between the activity of a neuronal population and a decision process is to bias decisions using electrical stimulation (or optogenetics), and it is true that no such link has yet been established between the activity of *offer value* cells in OFC and economic decisions. (Incidentally, establishing such link in this case is technically difficult due to the lack of columnar organization in OFC.) Thus it cannot be ruled out that economic decisions are based on entirely different populations of cells that *do not* adapt to the value range. In the remaining of this response, we refer to this scenario as the "alternative hypothesis".

The alternative hypothesis would explain the lack of choice bias in Exp.2. However, several lines of evidence argue against it.

First, clear evidence from clinical and lesion studies indicates that a well functioning OFC is necessary for economic decisions and goal-directed behavior. Indeed, OFC lesions result in choice deficits in various domains and in increased violations of choice transitivity (Camille et al., 2011; Fellows, 2004; Rahman et al., 2001). Furthermore, experiments using the reinforcer devaluation paradigm indicate that deficits following OFC lesions are due to the inability to compute values (Gallagher et al., 1999; Padoa-Schioppa and Schoenbaum, 2015; Rudebeck and Murray, 2011; Rudebeck et al., 2013; West et al., 2011). To reconcile the alternative hypothesis with this literature, one would have to imagine a scenario in which (a) OFC is necessary to compute values, (b) OFC is necessary to make economic decisions, but (c) OFC neurons

encoding offer values during economic decisions are not involved in the decision. Practically speaking, this scenario seems somewhat unlikely.

Second, previous work found a correlation between trial-by-trial fluctuations in the neuronal activity of *offer value* cells and the decision made on any given trial (Conen and Padoa-Schioppa, 2015; Padoa-Schioppa, 2013). For each cell, this correlation is quantified with an ROC analysis, which provides a choice probability (CP). Compared to the CPs measured in area MT during perceptual decisions (Britten et al., 1996; Britten et al., 1992), the CPs of *offer value* cells are relatively small. However, CPs are linearly related to noise correlations (Haefner et al., 2013; Shadlen and Newsome, 1998), and noise correlations in *offer value* cells are much lower than those found in area MT (Zohary et al., 1994). Simulations based on Haefner's equation showed that the noise correlations measured for *offer value* cells combined with a linear decision model closely reproduced the experimental measures of CPs (Conen and Padoa-Schioppa, 2015). In other words, given the level of noise correlation in OFC, the CPs of *offer value* cells are just as large as one would expect under the assumption that decisions are primarily based on their firing rates.

Third, as noted by Reviewer 2, the relation between choice variability and the value ranges observed in Exp.1 and Exp.2 argues against the alternative hypothesis described above (the Reviewer's comments on that relation are addressed below).

Lastly, it is not clear where in the brain one might find the putative non-range-adapting *offer value* cells assumed under the alternative hypothesis. Indeed, while value signals exist in numerous brain regions, most of the evidence outside OFC is about neurons encoding the *chosen value*, not the *offer value* (Amemori and Graybiel, 2012; Cai et al., 2011; Cai and Padoa-Schioppa, 2012, 2014; Lau and Glimcher, 2007; Roesch et al., 2009). One candidate might be the ventromedial prefrontal cortex (vmPFC). However, fMRI studies that focused on signals associated with individual goods or options (i.e., offer values) generally found such signals in OFC as opposed to vmPFC (Howard et al., 2015; Howard and Kahnt, 2017; Klein-Flugge et al., 2013). Another candidate might be the amygdala (Grabenhorst et al., 2016; Paton et al., 2006). However, neurons in the amygdala do undergo range adaptation (Bermudez and Schultz, 2010). Finally, one candidate might be the lateral intraparietal (LIP) area and other areas where neurons encode action values (Louie and Glimcher, 2010; Platt and Glimcher, 1999). However, it is generally understood that these areas receive offer value signals computed elsewhere (Cisek, 2012; Louie et al., 2014).

In conclusion, while causal links in the gold-standard sense discussed above have not yet been established, a broad spectrum of evidence supports the hypothesis that *offer value* cells in OFC provide a primary input for economic decisions. Indeed, most current neuro-computational models embrace this view (Cisek, 2012; Friedrich and Lengyel, 2016; Kable and Glimcher, 2009; Rustichini and Padoa-Schioppa, 2015; Solway and Botvinick, 2012; Song et al., 2017). (Note that these models include both action-based and good-based models.)

In the revised ms, we included a new paragraph in the Discussion (p.10) to discuss the important point raised by Reviewer 2:

The rationale for this study rests on the assumption that *offer value* cells in OFC provide the primary input for the neural circuit that generates economic decisions. Support for this assumption comes from lesion studies (Camille et al., 2011; Gallagher et al., 1999; Rudebeck and Murray, 2011), from the joint analysis of choice probability and noise correlation (Conen and Padoa-Schioppa, 2015) and from the relation between choice variability and value range shown here. Indeed, current neuro-computational models of economic decisions embrace this view (Cisek, 2012; Friedrich and Lengyel, 2016; Kable and Glimcher, 2009; Rustichini and Padoa-Schioppa, 2015; Song et al., 2017). However, we note that causal links between the activity of *offer value* cells and the decision have not yet been demonstrated with the gold-standard approach of biasing choices using electrical or optical stimulation. Future work should fill this important gap.

(2) Perhaps the most interesting and potentially important data from the paper is the finding that monkey choice behavior is value range-sensitive. Past papers have documented range adaptation in OFC neurons but no clear corresponding behavioral correlates (Padoa-Schioppa 2009; Kobayashi et al 2010). However, there is a critical point of analysis that confounds the argument that payoff (or the inverse of sigma) is related to the value range.

Figure 5 and the accompanying analyses aim to compare payoffs (or equivalently choice curve slopes, or sigma) across different sessions with different value ranges. However, in the data (from Padoa-Schioppa and Assad, 2006), there were ~12 juice types and individual sessions presented single pairs of these types. The fundamental issue: how can the behavior in different sessions be compared when different juices are used across sessions? Specifically, how are the x axes in Fig. 5 co-aligned for different sessions? Similarly, how can the sigmoid fits be compared across different sessions with different juice types?

Both Fig. 5 and the sigmoid fits are performed with each session analyzed in objective units of q_B (drops of juice B). But this assumes that a given change in juice B in one session is equivalent to a given change in juice B in a different session, in *subjective* terms. However, there is no reason to think that the subjective change in value of a given quantity change in grape juice is equivalent to that of the same objective quantity change in peppermint tea. One could argue that utilizing ratios of goods (in equivalent B units) as the authors do could alleviate the problem, but this approach assumes that the linear scaling of utility for separate goods are the same. If the x axes in Fig. 5a,b cannot be aligned across sessions, then it is impossible to compare payoff/choice stochasticity across different sessions.

Throughout the analysis we assume linear indifference curves (Fig.4a). If we indicate with q_X the quantity of juice X and with $V(q_X)$ its value, one can generally write $V(q_A) = q_A^\alpha$ and $V(q_B) = q_B^\beta$. Assuming linear indifference curves means assuming $\alpha = \beta$. Given that the quantities offered in our experiments are relatively small (0-10 drops), this assumption is relatively mild. In practice, and especially since the tuning of *offer value* cells is quasi-linear, it would seem safe to make the stronger assumption $\alpha = \beta = 1$. However, in generating Fig.5ab, we make a weaker assumption, namely $\alpha / \beta = \text{constant}$ across sessions. Under this assumption, the x-axis in Fig.5ab is proportional to the log value ratio. (It is equal to the log value ratio if $\alpha / \beta = 1$.)

Following on these considerations, Reviewer 2 is correct that the subjective value of juice B (likely) varies from session to session. However, using the value ratio (a pure number) makes it

possible to align sigmoid functions across sessions under relatively weak assumptions. Note also that Fig.5ab is mostly illustrative – the actual statistical analysis is done in Fig.6 (see below).

On a related note, I'm not sure that value ranges can be compared across sessions either. For example the correlational analyses in Fig. 5c,d rely on a measure of mean value range, defined as $(\rho * Q_A + Q_B) / 2$. This means that each session range is defined in objective units (drops) of a particular good (good B). I don't see how ranges across sessions can be compared when they are in different objective (different goods) and subjective (the utility of those goods, which are not compared) units.

As a clarification, the x-axis in Fig.6ab (formerly Fig.5cd) is a geometric mean, although the figure generated using the simple mean is almost identical (because $\rho Q_A \approx Q_B$ in the experiments). We clarified this point in the figure legend.

Apart from that, the point raised by Reviewer 2 is well taken, and we had not given this issue sufficient thought – so thank you! It is true that the subjective value of a unit quantity of juice B (uB) likely differs from session to session, and thus putting on the same axis value ranges from different sessions is tricky. We formalize this issue as follows. Lets say there was some good G such that the subjective value of a unit quantity of G (uG) remains fixed across sessions. We could express $uB_s = c_s uG$, where s indicates the session, and c_s is a proportionality factor. In building Fig.6ab we acted as though c_s was constant, but in reality c_s likely varies from session to session. However, the figure remains essentially valid if we assume that variations in the sigmoid width (σ) are independent of variations in c_s . This assumption is actually fairly conservative. If anything, we would expect that when c_s is smaller, the monkey is less motivated to work, and thus σ increases. If so, variability in c_s would induce a correlation in the direction opposite to that observed in the data.

To acknowledge this important issue, we added the following paragraph in the Methods (p.23):

Importantly, Δ in Fig.6ab is expressed in value units of juice B and the figure pools data from different sessions. However, the subjective value of a unit quantity of juice B (uB) likely differs from session to session, and thus pooling data must be done with caution. To formalize this point, imagine there was a good G such that the subjective value of a unit quantity of G (uG) remains fixed across sessions. We could express $uB_s = c_s uG$, where s indicates the session, and c_s is a proportionality factor. In building each panel of Fig.6, we essentially ignored the fact that c_s may vary from session to session. More precisely, we assumed that variations in the sigmoid width (σ) are independent of variations in c_s . This assumption seems fairly conservative. If anything, one might expect that when c_s is smaller, the monkey is less motivated to work, and thus σ increases. If so, variability in c_s would induce a correlation in the direction opposite to that observed in the data.

Given these caveats, the data from Expt. 2 (Fig. S5) *can* be analyzed because they address offer range changes without changing goods. The Supplementary Material provides some evidence that there is such a relationship in a limited set of the data. However, to make the argument that there is a relationship between payoff and value range, the authors would have to do a fuller analysis of all the data from the experiment (i.e. including the sessions where A changes).

We revised Fig.S6 (formerly Fig.S5) and included data from sessions in which Q_A was manipulated. When these sessions were considered alone, the change in sigmoid width (σ) was not significant. However, when all sessions were pooled together, the change in sigmoid width was statistically significant ($p < 0.05$, sign test; $p < 0.05$, t-test).

We are not completely sure of why the effect was not significant when we only manipulated Q_A . Our current thinking is as follows: Because juice A was always preferred, we generally used smaller quantities. In value space, this means that the values of juice A were sampled more coarsely. Furthermore, given our joint distributions of offers (see Fig.S7), animals split decisions only when juice A was offered in quantity 1. These two factors combined may have helped the animal and reduce the predicted behavioral effect. (This is why we originally excluded these sessions from the figure.)

(3) The discussion raises several issues that are currently unaddressed, and the authors should clear up for the reader. First, it's not clear to me which optimization goal is being served by range adaptation: the paper argues that decision model maximizes payoff at different value ranges, but also presents results that suggest that quasi-linearity mimics ORF for joint uniform reward distributions. In the former, optimization is for decision outcomes, whereas in the latter, optimization is - as for sensory systems - for input (reward) coding.

The tuning curves of *offer value* cells are quasi-linear and they are range adapting. The first question is: *Why range adaptation?* The answer is that, given linear tuning functions, range adaptation maximizes the expected payoff. The second question is: *Why quasi-linearity?* The answer here is more articulated. First, it is clear that quasi-linearity *does not* maximize information transmission in our experiments ($n_{\text{trialsCDF}}$ would, but see Fig.1), so there is no analogy with sensory systems. Second, for the purpose of maximizing the expected payoff, given the joint distributions of offers used in our experiments, quasi-linearity was not optimal (Fig.7b-d). This observation suggests that quasi-linearity is a property of *offer value* tuning functions that is not subject to contextual adaptation. If so, it is reasonable to ask: *Is there a joint distribution of offers for which quasi-linearity would be optimal?* The answer is that quasi-linearity would be optimal if the joint distribution was uniform (with equal value ranges). Thus, in layman terms, one might say that *offer value* cells implicitly "assume" that the joint distribution of offers is uniform with equal value ranges. The last question is: *Why would the economic decision circuit make such "assumption"?* Here the answer is admittedly speculative (this is a Discussion point):

Consider an *offer value B* cell, and assume strict menu invariance. In other words, the cell "sees" only the marginal distribution of B values and "knows nothing" about good A. In particular, the cell doesn't know how desirable A is (i.e., it doesn't know the relative value ρ) and it doesn't know the actual distribution of A values. The cell can "assume" that the joint distribution of offers is uniform with equal value ranges without having any knowledge of ρ or of the actual joint distribution. This is because the information about the range of B values is sufficient to make that assumption. (This addresses the next comment of Reviewer 2.) This line of reasoning motivates the speculation that tuning functions are quasi-linear because, by being menu invariant, they facilitate preference transitivity.

Second, the relationship between joint reward distribution-based response functions and menu independence (Discussion, pg 8-9) is puzzling to me. The authors note that OFC value coding is

menu invariant, and that each neuron adapts to its own value range, suggesting that “neurons associated with one particular good are blind to every aspect of the other good”. However, an ORF that is tuned to the *joint* uniform distribution of rewards must take into account the range of the other good. This seems like a contradiction to me, perhaps the authors can clear this up?

This issue is addressed in the second part of the previous response. Basically, knowing one of the two value ranges and nothing else is sufficient to define a joint distribution of offer values that is uniform with equal value ranges. Thus a neuron that "sees" only one juice can be optimally tuned for such a uniform distribution.

Other points

(1) The abstract stated that “we show that neuronal responses are quasi-linear even when optimal tuning functions would be highly non-linear.”. It’s crucial here that the authors state what is meant by optimal: is it optimal in the classic sensory CDF sense, or optimal in terms of expected choice payoff?

In the context of economic decisions, optimality is always meant in the sense of the expected payoff. We have limited space to clarify this point in the Abstract, but we do so in the Introduction and in a new, dedicated section (see above, *Introductory remarks on this revision*).

(2) Clarity of the model and theoretical predictions. In the synopsis of model results (pg 5-6), the main point is that maximal expected payoff occurs when there is complete range adaptation. In other words, given a range of offer values, maximum payoff occurs with complete adaptation of firing rates. However, the prediction in the next section that payoff and value range are inversely related is not addressed and it is not clear from where it arises.

Thanks for pointing out this issue. In essence, the probability of choosing juice A (Eq.2 in the main text) is the sigmoid function that represents choices. The width of the sigmoid is captured by its slope on the indifference line, which we can compute explicitly (Eq.28 in the Supplementary Material). In the revision, we added a short paragraph to discuss this point (p.7):

Notably, Eq.2 expresses the sigmoid surface describing choices. By computing the slope of this surface on the indifference line, we show that under optimal coding the sigmoid width is directly related to the value ranges (see Supplementary Material, Eq.28).

(3) How does a nonlinear value function of quantity (drops) affect these interpretations? One of the points of the paper is that there is a functional rigidity to OFC responses: linear coding of offered rewards (in drops). However, much work in the economics literature suggests that people and animals have nonlinear utility functions. One wonders if the linearity in response to rewards here is an experimental byproduct of the quantization of juice delivery in drops; if juices were delivered as different volumes rather than different numbers of drops, perhaps value coding would be nonlinear? I certainly don’t expect the authors to provide different experimental data, but could they address the relationship between linear value coding and nonlinear utility functions?

This is a very interesting question, for which we don't have a definite answer. However, we can make a few considerations.

In the economics literature, statements about the non-linearity of utility functions come mostly from observations of choices under risk. Human subjects are usually described as risk averse, while most experiments in neurophysiology settings found that rhesus macaques are risk seeking. Taking seriously the curvature observed in our *offer value* cells – in particular, the quadratic term β_2 (Fig.2a) – one would predict that these monkeys would be risk seeking. This is indeed what we observed in a previous study (Raghuraman and Padoa-Schioppa, 2014). In another study, Stauffer et al. (2014) found that monkeys are risk seeking for low values and risk averse for higher values. If so, and if risk attitudes are ultimately due to the curvature of the tuning functions of *offer value* cells, then one would expect to observe slightly S shaped tuning in these neurons. Interestingly, our data conform to this prediction (Fig.2b). We should also note that the theory presented in the SM accounts for the quadratic term (the ORF in Fig.6e is convex). However, in our theory ORF does not depend on the value range, provided that the two ranges are the same (symmetric case). Thus this theory does not account for risk attitudes in general.

Are the results observed here a byproduct of quantization? Technically, juices in Exp.1 were delivered to the animals in continuous boluses (i.e., the solenoid opened and closed only once), although we did play brief sounds according to the number of drops (e.g., the same sound was repeated 6 times when the animal was delivered 6 "drops" of juice). In later experiments, we changed this aspect of the experimental design (i.e., we opened and closed the solenoid for each drop of juice chosen by the monkey) and we did not notice any difference. Our hunch is that the tuning curvature of *offer value* cells does not depend on quantization per se.

(4) How rho (relative value) is defined is not clear from the text. I understand that it is quantified as the slope of the indifference line (Fig. 4a) between offer A and offer B, and essentially measures the tradeoff between objective amounts of A and B, but it should be clarified in the text when first discussed (pg 5?).

We clarified this point on p.4 (and added Fig.1b) as follows:

In each session, the choice pattern was fitted with a sigmoid function, and the flex of the sigmoid provided a measure for the relative value of the two juices, referred to as ρ (Methods, Eq.5). The relative value allows to express quantities of the two juices on a common value scale. In one representative session, we measured $\rho = 4.1$ (Fig.1b).

(5) Can the authors clarify what they mean by "payoff" in the main text? It is never clearly defined, but presumably this is a measure of how often the animal chooses the higher subjectively valuable reward (which itself is defined by the animal's behavior in the block).

See above, *Introductory remarks on this revision*. The expected payoff is defined as the chosen value averaged across trials.

(6) Figure 5. Can the authors clarify what the blue-green shading depicts? Does it represent the size of the mean value range? Please either state explicitly in the legend, or provide a visual colorbar.

Different shades of color (from blue to green) indicate the ordinal ranking of sessions according to the mean value range (Δ). We clarified this point in the figure legend.

(7) Figure 6e. The legend describes the case where $\rho=2$, but the dashed line in panel (e) depicts $\rho=1$.

We corrected the figure -- thanks!

(8) Optimal response functions (Fig 6). Not sure how they are derived. The idea that sensory neural tuning curves approximate the CDF of the encoded statistic makes specific certain assumptions, such as the desire for a uniform utilization of output FRs (maximizing entropy/information). Can the authors be clearer about how the ORFs are derived?

We include in the submission the Matlab script used to compute the ORF. In words, the program estimates the optimal response function (ORF) by setting the domain of the function as a discrete grid of dimension G , and representing the RF as a vector. The function to be maximized is the expected payoff, calculated as the probability of choosing the good A times the value of good A, plus the probability of choosing the good B times the value of good B. The RF is constrained to take values in the unit interval $[0,1]$. The program then implements the non-linear constrained maximization of the expected payoff under the constraints. Parameters are chosen as described in the main text, to be consistent with the physiological values identified there.

(9) In the presentation summarizing the model implications (pg 6), the authors discuss symmetric and asymmetric cases, but it would help the reader if the explicit difference between the two are stated in plain terms. I believe symmetric describes instances when the range of the two goods are equivalent (in good B-equivalent units), and asymmetric describes when they are not.

The understanding of Reviewer 2 is correct, and we clarified this point on p.7 as follows:

In the symmetric case, defined by $\rho Q_A = Q_B$ (equal value ranges), the expected payoff is maximal [...]. In the asymmetric case, (unequal value ranges), there is a small choice bias [...].

Reviewer #3 (Remarks to the Author):

In this manuscript, Rustichini et al. present deeper investigations of data and results related to a series of publications from the Padoa-Schioppa lab, in which orbitofrontal (OFC) neurons in monkeys are found to encode decision parameters in choices involving quantity-quality trade-offs. The authors conducted a series of analyses collectively aimed at elucidating the nature of value coding in OFC neurons, particularly in the context of range adaptation.

The previous studies from this group report three coding schemes common among OFC neurons (offer value, chosen value, chosen juice), and the present manuscript focuses on those encoding 'offer value', referring to the value of one of two juice rewards available in a given task session. These neurons are believed to encode pre-decision variables, such that a choice can be made by comparing the relative firing rates of two 'offer value' populations coding for each option. It has previously been reported that they exhibit range adaptation. That is, their firing rates do not faithfully reflect the absolute quantity of juice, but rather adapt so that baseline and maximal firing rates roughly map onto the minimum and maximum quantities available in a given session, with intermediate values linearly related.

Given this background, the authors first show that firing rates of offer value neurons adapt based on the range of values available in a session, but not the frequency with which these values are encountered (i.e. their firing rates are "quasi-linear" with respect to value). This result is novel if not entirely surprising. Indeed, it could provide interesting support for recent ideas that OFC neurons form "cognitive state space maps", in that the neurons appear to respond based on stored knowledge of the range of possible values (i.e. 'maps') rather than being influenced by the statistics of individual trials.

This result is loosely tied to the final part of the paper, in which the authors reanalyzed their data to assess non-linear explanations of firing rates, none of which outperformed the linear models. Only explanatory variables based on uniform joint distributions of the two offered values performed comparably to the linear models. However, whether a joint distribution of offer values provides a better description of neuron firing is not precisely resolved here, as both models fit the neural data well and were not statistically distinguishable.

A second theme of the paper was investigating an apparent paradox of adaptive coding among these neurons, which can be paraphrased as follows: if offer value neurons adapt their firing rates based on the available value range of one juice, and if we assume that the choice is a result of a comparison of offer value A vs. offer value B firing rates, then the indifference point in choices would shift depending on the range of values available. Because this shift is not observed behaviorally, it might be concluded that the choice is not strictly based on the relative firing rates of offer value neurons. This is an interesting observation. To drive this home, the authors tested monkeys in a nearly identical task, in which daily sessions were blocked so that the range of values in block 2 either increased or decreased. They showed that, behaviorally, the indifference point did not shift with different value ranges, implying there must be a correction somewhere in the decision circuitry. The authors hypothesize a downstream correction mechanism. While this explanation is speculative and not entirely satisfying, I think it is beyond the scope of this paper to find and demonstrate such a corrective mechanism.

The final piece of the story was a demonstration that range adaptation maximized expected payout.

We thank Reviewer 3 for the overall evaluation and for the specific comments.

[A] This argument was difficult to follow, in part because much of it was relegated to supplementary material, and also because it was not immediately clear how the authors defined and quantified ‘payoff’. I gather that it’s based on the value obtained as an abstract quantity not tied to the actual juice (i.e. ‘chosen value’?), but this isn’t really spelled out anywhere. [B] If this interpretation is correct, shouldn’t it be closely tied to the maximum value available in a session, not just the value range?

[Square brackets are ours]

[A] We apologize for the confusion, and we extensively revised the ms to clarify the concept of expected payoff. Please see above the *Introductory remarks on this revision*.

[B] For each juice, the value range is defined as the difference between the *max value* and the *min value*, the latter of which is always zero in these experiments. So *value range* is indeed equal to *max value*. In this respect, the Reviewer is correct. A different question is whether neurons adapt to the max value across juices. For example lets express all values in the same units and lets say that *max value A* = 8 and *max value B* = 10. In principle, *offer value A* cells might adapt to the global max, namely 10. This would actually solve the problem raised in the Introduction, in the sense that there would be no choice bias. However, we previously showed that this is not the case, and that each neuron adapts to *its own* value range (Padoa-Schioppa and Rustichini, 2014).

Further comments

(1) Overall I found this a challenging paper. There are interesting investigations here, but the manuscript as written relies heavily on self-referential ideas and terms making it difficult for the reader to parse what exactly was hypothesized and how it was addressed, even more so if they do not specialize in the area. It would be significantly improved by making it more accessible to a general audience.

We are sorry that the ms came across as challenging. In the revision, we tried to simplify the exposure as much as possible. Also, introducing and clarifying the concept of expected payoff early on should make the paper much easier to read.

(2) The authors seem to expect that these neurons should behave in a manner that is optimized for performing binary choices, for example in assuming that the optimal coding strategy is a step function that relates to the value of one juice option, with the step at the indifference point. This would, indeed, be optimal in the case of a familiar 2-option choice. But in how many other situations would this be optimal? It seems that having more parametric information, as provided by a response that varies more linearly with value (as was actually observed in the neurons), would be optimal for a wider range of behaviors. So in effect, is the argument that the neurons “should” respond via a step function something of a straw man?

This is an interesting observation, and it is very much in line with our understanding. The bottom line of this study is that value coding in OFC is functionally rigid (i.e., neuronal responses are quasi-linear independent of the behavioral context) but parametrically plastic (the gain adapts optimally). Why is there functional rigidity? We speculate that it facilitates choice transitivity. As Reviewer 3 suggests, it is also possible that quasi-linearity in *offer value* cells is optimal for other behaviors. However, the role of these neurons in other behaviors has not yet been examined. Furthermore, since our present focus is on binary economic decisions we think that bringing up other behaviors in this paper would ultimately confuse the reader. For these reasons we chose not to raise this point in the Discussion.

References

- Amemori, K., and Graybiel, A.M. (2012). Localized microstimulation of primate pregenual cingulate cortex induces negative decision-making. *Nat Neurosci* *15*, 776-785.
- Barlow, H.B. (1961). Possible principles underlying the transformations of sensory messages. In *Sensory Communication*, W.A. Rosenblith, ed. (Cambridge, MA: MIT Press), pp. 217–234.
- Bermudez, M.A., and Schultz, W. (2010). Reward magnitude coding in primate amygdala neurons. *J Neurophysiol* *104*, 3424-3432.
- Brenner, N., Bialek, W., and de Ruyter van Steveninck, R. (2000). Adaptive rescaling maximizes information transmission. *Neuron* *26*, 695-702.
- Britten, K.H., Newsome, W.T., Shadlen, M.N., Celebrini, S., and Movshon, J.A. (1996). A relationship between behavioral choice and the visual responses of neurons in macaque MT. *Vis Neurosci* *13*, 87-100.
- Britten, K.H., Shadlen, M.N., Newsome, W.T., and Movshon, J.A. (1992). The analysis of visual motion: a comparison of neuronal and psychophysical performance. *J Neurosci* *12*, 4745-4765.
- Cai, X., Kim, S., and Lee, D. (2011). Heterogeneous coding of temporally discounted values in the dorsal and ventral striatum during intertemporal choice. *Neuron* *69*, 170-182.
- Cai, X., and Padoa-Schioppa, C. (2012). Neuronal encoding of subjective value in dorsal and ventral anterior cingulate cortex. *J Neurosci* *32*, 3791-3808.
- Cai, X., and Padoa-Schioppa, C. (2014). Contributions of orbitofrontal and lateral prefrontal cortices to economic choice and the good-to-action transformation. *Neuron* *81*, 1140-1151.
- Camille, N., Griffiths, C.A., Vo, K., Fellows, L.K., and Kable, J.W. (2011). Ventromedial frontal lobe damage disrupts value maximization in humans. *J Neurosci* *31*, 7527-7532.
- Cisek, P. (2012). Making decisions through a distributed consensus. *Curr Opin Neurobiol*.
- Conen, K.E., and Padoa-Schioppa, C. (2015). Neuronal variability in orbitofrontal cortex during economic decisions. *J Neurophysiol* *114*, 1367-1381.
- Fellows, L.K. (2004). The cognitive neuroscience of human decision making: a review and conceptual framework. *Behav Cogn Neurosci Rev* *3*, 159-172.
- Friedrich, J., and Lengyel, M. (2016). Goal-directed decision making with spiking neurons. *J Neurosci* *36*, 1529-1546.
- Gallagher, M., McMahan, R.W., and Schoenbaum, G. (1999). Orbitofrontal cortex and representation of incentive value in associative learning. *J Neurosci* *19*, 6610-6614.

- Grabenhorst, F., Hernadi, I., and Schultz, W. (2016). Primate amygdala neurons evaluate the progress of self-defined economic choice sequences. *Elife* 5.
- Haefner, R.M., Gerwinn, S., Macke, J.H., and Bethge, M. (2013). Inferring decoding strategies from choice probabilities in the presence of correlated variability. *Nat Neurosci* 16, 235-242.
- Howard, J.D., Gottfried, J.A., Tobler, P.N., and Kahnt, T. (2015). Identity-specific coding of future rewards in the human orbitofrontal cortex. *Proc Natl Acad Sci U S A*.
- Howard, J.D., and Kahnt, T. (2017). Identity-specific reward representations in orbitofrontal cortex are modulated by selective devaluation. *J Neurosci* 37, 2627-2638.
- Kable, J.W., and Glimcher, P.W. (2009). The neurobiology of decision: consensus and controversy. *Neuron* 63, 733-745.
- Klein-Flugge, M.C., Barron, H.C., Brodersen, K.H., Dolan, R.J., and Behrens, T.E. (2013). Segregated encoding of reward-identity and stimulus-reward associations in human orbitofrontal cortex. *J Neurosci* 33, 3202-3211.
- Lau, B., and Glimcher, P.W. (2007). Action and outcome encoding in the primate caudate nucleus. *J Neurosci* 27, 14502-14514.
- Laughlin, S. (1981). A simple coding procedure enhances a neuron's information capacity. *Z Naturforsch C* 36, 910-912.
- Louie, K., and Glimcher, P.W. (2010). Separating value from choice: delay discounting activity in the lateral intraparietal area. *J Neurosci* 30, 5498-5507.
- Louie, K., LoFaro, T., Webb, R., and Glimcher, P.W. (2014). Dynamic divisive normalization predicts time-varying value coding in decision-related circuits. *J Neurosci* 34, 16046-16057.
- Padoa-Schioppa, C. (2013). Neuronal origins of choice variability in economic decisions. *Neuron* 80, 1322-1336.
- Padoa-Schioppa, C., and Rustichini, A. (2014). Rational attention and adaptive coding: a puzzle and a solution. *American Economic Review: Papers and Proceedings* 104, 507-513.
- Padoa-Schioppa, C., and Schoenbaum, G. (2015). Dialogue on economic choice, learning theory, and neuronal representations. *Current opinion in behavioral sciences* 5, 16-23.
- Paton, J.J., Belova, M.A., Morrison, S.E., and Salzman, C.D. (2006). The primate amygdala represents the positive and negative value of visual stimuli during learning. *Nature* 439, 865-870.
- Platt, M.L., and Glimcher, P.W. (1999). Neural correlates of decision variables in parietal cortex. *Nature* 400, 233-238.
- Raghuraman, A.P., and Padoa-Schioppa, C. (2014). Integration of multiple determinants in the neuronal computation of economic values. *J Neurosci* 34, 11583-11603.

- Rahman, S., Sahakian, B.J., Cardinal, R.N., Rogers, R., and Robbins, T. (2001). Decision making and neuropsychiatry. *Trends Cogn Sci* 5, 271-277.
- Roesch, M.R., Singh, T., Brown, P.L., Mullins, S.E., and Schoenbaum, G. (2009). Ventral striatal neurons encode the value of the chosen action in rats deciding between differently delayed or sized rewards. *J Neurosci* 29, 13365-13376.
- Rudebeck, P.H., and Murray, E.A. (2011). Dissociable effects of subtotal lesions within the macaque orbital prefrontal cortex on reward-guided behavior. *J Neurosci* 31, 10569-10578.
- Rudebeck, P.H., Saunders, R.C., Prescott, A.T., Chau, L.S., and Murray, E.A. (2013). Prefrontal mechanisms of behavioral flexibility, emotion regulation and value updating. *Nat Neurosci* 16, 1140-1145.
- Rustichini, A., and Padoa-Schioppa, C. (2015). A neuro-computational model of economic decisions. *J Neurophysiol* 114, 1382-1398.
- Shadlen, M.N., Britten, K.H., Newsome, W.T., and Movshon, J.A. (1996). A computational analysis of the relationship between neuronal and behavioral responses to visual motion. *J Neurosci* 16, 1486-1510.
- Shadlen, M.N., and Newsome, W.T. (1998). The variable discharge of cortical neurons: implications for connectivity, computation, and information coding. *J Neurosci* 18, 3870-3896.
- Simoncelli, E.P., and Olshausen, B.A. (2001). Natural image statistics and neural representation. *Annu Rev Neurosci* 24, 1193-1216.
- Solway, A., and Botvinick, M.M. (2012). Goal-directed decision making as probabilistic inference: a computational framework and potential neural correlates. *Psychological Review* 119, 120-154.
- Song, H.F., Yang, G.R., and Wang, X.J. (2017). Reward-based training of recurrent neural networks for cognitive and value-based tasks. *Elife* 6.
- Stauffer, W.R., Lak, A., and Schultz, W. (2014). Dopamine reward prediction error responses reflect marginal utility. *Curr Biol* 24, 2491-2500.
- West, E.A., DesJardin, J.T., Gale, K., and Malkova, L. (2011). Transient inactivation of orbitofrontal cortex blocks reinforcer devaluation in macaques. *J Neurosci* 31, 15128-15135.
- Zohary, E., Shadlen, M.N., and Newsome, W.T. (1994). Correlated neuronal discharge rate and its implications for psychophysical performance. *Nature* 370, 140-143.

Reviewers' comments:

Reviewer #1 (Remarks to the Author):

The authors have addressed all of my concerns. I have no further comments.

Reviewer #2 (Remarks to the Author):

Rustichini et al present a revised manuscript with theoretical/empirical work examining value adaptation in OFC neurons, the functional form of value coding, and the relationship between value coding, adaptation, and payoff maximization. Their revisions nicely address a number of previous concerns, and have improved the manuscript. I appreciate and accept their arguments re: the putative causal role of OFC value coding in the choice process, and the added points in the discussion help clear up the authors' viewpoint.

However, I still have some concerns about the experimental work examining the relationship between choice variability and value range. I realize that demonstrating the variability-range relationship is not the central point of the paper, but there are some points the authors should address. My concerns are threefold:

(1) Correctness of aligning value ranges and relative value axes across sessions (different juices, different days). I still am unsure that the authors can reasonably compare across sessions in their primary dataset (Expt 1), where the juice pairings are different from day to day. In their reply, the authors point out that the log value ratio scale (Fig. 5) is equivalent to assuming that the alpha/beta ratio is constant across days (where $V(qA) = qA^\alpha$ and $V(qB) = qB^\beta$). While this is true, the experimental setup is such that different juices are paired on different days - meaning, I presume, that a juice could be juice A on one day and juice B on another. This makes assuming a constant alpha/beta ratio a little less well founded.

This is also an issue for comparing value ranges across days with different juice pairs. Consider an example where day 1 pairs orange and apple (orange preferred to apple) and day 2 pairs apple and grape (apple preferred to grape). If the same range of apple is given each day (10 drops), it will contribute $QB=10$ to the value range on day 1 but $\rho*QA=30$ (if $\rho=3$) on day 2. The problem is that everything is scaled to the non-preferred juice, but the identities of juices are not the same across sessions.

In their reply, the authors point out that you can think of a unit quantity of some good G (uG) as having a variable value session-to-session where $uB_s = c_s * uG$, and that c_s is likely independent of variability. However, this approach is for a given good - the real issue is that different goods are used in different sessions, and there is no reason to expect $uB(\text{apple}) = uB(\text{grape})$ [the value of 1 drop of apple has no relationship to the value of 1 drop of grape]. The authors rightly point out that they *do* observe a relationship between sigma and value range, and that it is only an issue if c_s (or the scaling of different goods, but same idea) varies with sigma. This is a difficult assumption to address in the data, so perhaps a more important point to address is the following: is there another potential explanation for the relationship between sigma and value range? This is my next point below.

(2) Controlling for other explanations for the range-variability relationship. In looking at the definition of the logistic fit parameters (rho and sigma), it is clear that they both depend on the same parameter (a1). The issue of concern is that the relationship the authors wish to show (Fig. 6) is a relationship between choice variability and value range; choice variability is defined as sigma, which has some

relationship to rho (they both depend on a1), and value range also depends on rho ($[\rho \cdot Q_a \cdot Q_b]^{.5}$).

So the key question is: is there a relationship between rho and sigma? In other words, do the animals show greater variability in sessions where the relative values of the goods are more disparate? A simple way to check is simply to see if there is a correlation between the quantified rho and sigma parameters across sessions. If there is a relationship, then I am concerned that it is driving the apparent relationship between sigma and value range (since both would be a function of rho). Even if there is a relationship between sigma and rho, the authors could examine the variability-range relationship while controlling for rho (either by regression or by stratifying by rho).

(3) Within day range-variability relationship (Expt. 2). As discussed in the last round of reviews, the data in Expt. 2 offer a way around the issues of comparing across different goods, since the comparisons are made within day. However, it is a bit concerning that all the data don't show the intended effect (Fig. S6). Do the authors have any idea why there is such a marked difference between the A and B manipulations?

Other points

(1) In Fig. 4A, the authors show the indifference line for goods A and B, and refer to the slope of the line as rho (the relative value, defined as the number of B units equivalent to 1 unit of A). I think that - as plotted - the slope of the line is actually $1/\rho$ (since the plot depicts A vs B). By definition, the line can only get shallower than the unity line, but rho is ≥ 1 .

(2) Minor detail re: Fig. 3A. The legend describes the blue line as depicting rho = 3, but the actual figure shows a rho ~2.5.

(3) Another suggestion for Fig. 3: the intuition outlined in 3A-B is difficult to apply to 3C due to the way the authors chose to denote their value range ratios. In 3A-B, the authors show that increasing the range of B should (in the uncorrected case) increase the relative value of A:B. However, the axes in C show $\Delta A/\Delta B$ and plot the ratio=X case vs. ratio=2X. Seems to me that it would be easier for the reader if the graph were labelled as $\Delta B/\Delta A$, with the x-axis being "X" and the y-axis being "2X". I realize these are equivalent, but it would match the intuition depicted in 3A-B.

(4) I'm not sure I understand the necessity of Fig. S5. Isn't the inverse temperature just a transformed version of sigma?

Reviewer #3 (Remarks to the Author):

I have reviewed the revised manuscript. My main concern in the previous version was that it was difficult to follow. I think that the authors' revisions and additional comments have adequately addressed these concerns. In particular, fleshing out the idea of expected payoff as it relates to the preference function helped significantly. My only comment is that there is a typo in the first paragraph of the Results ("allows to express"). I have no other significant concerns.

Reviewer #2 (Remarks to the Author):

Rustichini et al present a revised manuscript with theoretical/empirical work examining value adaptation in OFC neurons, the functional form of value coding, and the relationship between value coding, adaptation, and payoff maximization. Their revisions nicely address a number of previous concerns, and have improved the manuscript. I appreciate and accept their arguments re: the putative causal role of OFC value coding in the choice process, and the added points in the discussion help clear up the authors' viewpoint.

However, I still have some concerns about the experimental work examining the relationship between choice variability and value range. I realize that demonstrating the variability-range relationship is not the central point of the paper, but there are some points the authors should address.

We thank Reviewer 2 for carefully reading our ms and for the additional comments. We appreciate the concerns raised here, and we recognize that the relation between choice variability and value range deserved more attention. In this revision, we conducted new analyses that address all the key points. We also revised the ms as suggested by the Reviewer, and included other relatively minor corrections.

My concerns are threefold:

(1) Correctness of aligning value ranges and relative value axes across sessions (different juices, different days). I still am unsure that the authors can reasonably compare across sessions in their primary dataset (Expt 1), where the juice pairings are different from day to day. In their reply, the authors point out that the log value ratio scale (Fig. 5) is equivalent to assuming that the alpha/beta ratio is constant across days (where $V(qA) = qA^\alpha$ and $V(qB) = qB^\beta$). While this is true, the experimental setup is such that different juices are paired on different days - meaning, I presume, that a juice could be juice A on one day and juice B on another. This makes assuming a constant alpha/beta ratio a little less well founded.

This is also an issue for comparing value ranges across days with different juice pairs. Consider an example where day 1 pairs orange and apple (orange preferred to apple) and day 2 pairs apple and grape (apple preferred to grape). If the same range of apple is given each day (10 drops), it will contribute $QB=10$ to the value range on day 1 but $\rho*QA=30$ (if $\rho=3$) on day 2. The problem is that everything is scaled to the non-preferred juice, but the identities of juices are not the same across sessions.

In their reply, the authors point out that you can think of a unit quantity of some good G (uG) as having a variable value session-to-session where $uB_s = c_s * uG$, and that c_s is likely independent of variability. However, this approach is for a given good - the real issue is that different goods are used in different sessions, and there is no reason to expect $uB(\text{apple}) = uB(\text{grape})$ [the value of 1 drop of apple has no relationship to the value of 1 drop of grape]. The authors rightly point out that they do observe a relationship between sigma and value range, and that it is only an issue if c_s (or the scaling of different goods, but same idea) varies with sigma. This is a difficult assumption to address in the data, so perhaps a more important point to

address is the following: is there another potential explanation for the relationship between sigma and value range? This is my next point below.

(2) Controlling for other explanations for the range-variability relationship. In looking at the definition of the logistic fit parameters (rho and sigma), it is clear that they both depend on the same parameter (a1). The issue of concern is that the relationship the authors wish to show (Fig. 6) is a relationship between choice variability and value range; choice variability is defined as sigma, which has some relationship to rho (they both depend on a1), and value range also depends on rho ($[\rho \cdot Q_a \cdot Q_b]^{\Delta}$).

So the key question is: is there a relationship between rho and sigma? In other words, do the animals show greater variability in sessions where the relative values of the goods are more disparate? A simple way to check is simply to see if there is a correlation between the quantified rho and sigma parameters across sessions. If there is a relationship, then I am concerned that it is driving the apparent relationship between sigma and value range (since both would be a function of rho). Even if there is a relationship between sigma and rho, the authors could examine the variability-range relationship while controlling for rho (either by regression or by stratifying by rho).

Before addressing the most important issues, let us call attention to a specific point. The choice pattern measured in any session is fitted with a sigmoid function:

$$P_{ch=B} = \int_{-\infty}^X N(0,1) dt$$

$$X = a_0 + a_1 \log(q_B / q_A)$$

From the fitted parameters we compute the relative value $\rho = \exp(-a_0/a_1)$, which captures the indifference point. In turn, choice variability may be quantified in several ways. In the original ms, we presented most of the results in terms of the sigmoid width, defined as $\sigma = \exp(1/a_1)$. However, some of the analyses were more naturally presented in terms of the sigmoid steepness, defined as $\eta = a_1$ (a_1 is also referred to as "inverse temperature"). Partly to address a comment of Reviewer 2 (see below), in this revision we found it preferable to analyze all the data in terms of the sigmoid steepness. Thus throughout the ms we now discuss choice variability in terms of the sigmoid steepness.

With this premise, let us summarize the two main concerns of Reviewer 2 as follows. Fig.6 demonstrates the relation between choice variability and value range. More precisely, we illustrate a negative correlation between the sigmoid steepness (η) and the mean value range (Δ). Reviewer 2 correctly pointed out that this correlation might not be direct, and might instead reflect other sources of variability. There are two specific issues:

(1) Fig.6 includes sessions with different juice pairs, with different typical values of ρ . In principle, choice variability could vary from juice pair to juice pair in a way that induces the relation observed in Fig.6.

(2) For any given juice pair, sigmoid steepness (η), relative value (ρ) and value range (Δ) are all inter-related by definition (main text, Eq.6) and because in the experiments we often chose value ranges not independently of ρ (in many sessions we set $Q_A = 3$ and chose Q_B such that $Q_B \approx \rho Q_A$). Thus the relation between η and Δ (Fig.6) might reflect fluctuations in ρ .

To address these concerns, we divided sessions in different sets based on the animal and on the juice pair. We only considered sets with at least 5 sessions, and our data included 12 such sets (6 sets from each monkey). We then analyzed each set of sessions separately. We addressed the two issues as follows:

(1) We examined the relation between Δ and η for individual juice pairs. Confirming the population results, we found that the negative correlation between η and Δ was present in each of the 12 data sets. This result is illustrated in Fig.S5 and summarized in Fig.S6a .

(2) To assess whether the relation between Δ and η simply reflected fluctuations in ρ , we took the advice of Reviewer 2 and used multilinear regression. For each set of sessions, we regressed η on ρ and then on Δ in a stepwise way. The coefficient obtained from the second regression (β_η) quantified the correlation between η and Δ not explained by fluctuations of ρ . For most sets, we found $\beta_\eta < 0$, indicating that the residual η after regressing on ρ was still negatively correlated with Δ . Considering the 12 sets, the distribution of β_η was significantly displaced from zero (mean(β_η) = -0.31, p = 0.01, one-tailed t test). In other words, the negative correlation between η and Δ was above and beyond the correlation explained by fluctuations of ρ . The results of this analysis are illustrated in Fig.S6a.

We believe that these new results fully address the concerns of Reviewer 2. The revised ms discusses these issues in the Methods (p.26) and includes two new figures (Fig.S5 and Fig.S6).

(3) Within day range-variability relationship (Expt. 2). As discussed in the last round of reviews, the data in Expt. 2 offer a way around the issues of comparing across different goods, since the comparisons are made within day. However, it is a bit concerning that all the data don't show the intended effect (Fig. S6). Do the authors have any idea why there is such a marked difference between the A and B manipulations?

As we wrote in our previous response, we are not completely sure of why the effect was not significant when we only manipulated Q_A . One possibility is as follows: Because juice A was always preferred, we generally offered smaller quantities of A (typically 0, 1, 2, 3; see Fig.1). In value space, this means that offer values for juice A were sampled more coarsely. Furthermore, given our joint distributions of offers (Fig.S8), in practice animals split decisions only when juice A was offered in quantity 1. These two factors combined may have helped the animal and reduce the predicted effect on choice variability.

Other points

(1) In Fig. 4A, the authors show the indifference line for goods A and B, and refer to the slope of the line as rho (the relative value, defines as the number of B units equivalent to 1 unit of A). I think that - as plotted - the slope of the line is actually 1/rho (since the plot depicts A vs B). By definition, the line can only get shallower than the unity line, but rho is ≥ 1 .

Good point -- thanks. We corrected the figure.

(2) Minor detail re: Fig. 3A. The legend describes the blue line as depicting $\rho = 3$, but the actual figure shows a $\rho \sim 2.5$.

Good point -- thanks. We corrected the legend.

(3) Another suggestion for Fig. 3: the intuition outlined in 3A-B is difficult to apply to 3C due to the way the authors chose to denote their value range ratios. In 3A-B, the authors show that increasing the range of B should (in the uncorrected case) increase the relative value of A:B. However, the axes in C show $\Delta A/\Delta B$ and plot the ratio=X case vs. ratio=2X. Seems to me that it would be easier for the reader if the graph were labelled as $\Delta B/\Delta A$, with the x-axis being "X" and the y-axis being "2X". I realize these are equivalent, but it would match the intuition depicted in 3A-B.

We inverted the axes in the figure.

(4) I'm not sure I understand the necessity of Fig. S5. Isn't the inverse temperature just a transformed version of sigma?

We agree with Reviewer 2. In the revised ms we discuss all the results in terms of the inverse temperature (referred to as "sigmoid steepness" for clarity).

REVIEWERS' COMMENTS:

Reviewer #2 (Remarks to the Author):

The authors have done a thorough job of addressing my past concerns, and I think the paper merits publication.